# LEGO-EVAL: TOWARDS FINE-GRAINED EVALUATION ON SYNTHESIZING 3D EMBODIED ENVIRONMENTS WITH TOOL AUGMENTATION

## ABSTRACT

Despite recent progress in using Large Language Models (LLMs) for automatically generating 3D scenes, generated scenes often lack realistic spatial layouts and object attributes found in real-world environments. As this problem stems from insufficiently detailed, coarse-grained instructions, advancing 3D scene synthesis guided by more detailed, fine-grained instructions that reflect real-world environments becomes crucial. Without such realistic scenes, training embodied agents in unrealistic environments can lead them to learn priors that diverge significantly from real-world physics and semantics, degrading their performance when deployed. Thus, verifying the alignment between the fine-grained instruction and the generated scene is essential for effective learning. However, current evaluation methods, such as CLIPScore and vision-language models (VLMs), often fail to reliably assess such alignment. This shortcoming arises primarily from their shallow understanding of 3D scenes, which often leads to improperly grounded scene components. To address this, we introduce LEGO-EVAL, an evaluation framework equipped with diverse tools designed to explicitly ground scene components, enabling more accurate alignment assessments. We also present LEGO-BENCH, a benchmark of detailed instructions that specify complex layouts and attributes of real-world environments. Experiments demonstrate that LEGO-EVAL outperforms VLM-as-a-judge by 0.41 F1 score in assessing scene-instruction alignment. Benchmarking with LEGO-BENCH reveals significant limitations in current generation methods. Across all evaluated approaches, success rates reached at most 10% in generating scenes that fully align with fine-grained instructions.

## 1 INTRODUCTION

Embodied agents represent a paradigm shift from digital assistants to physical collaborators (Shridhar et al., 2020; Brohan et al., 2022; Chang et al., 2025). While training agents in the physical world is feasible, it is impractical due to the slow pace of real-time learning and the escalating costs of scaling across multiple environments. Consequently, training in realistic simulators has become the dominant approach, allowing agents to learn real-world physics and semantics through navigation and interaction in 3D scenes (Xiang et al., 2020; Makoviychuk et al., 2021; Li et al., 2022). Training in realistic scenes is critical, as unrealistic scenes can prevent agents from learning the physical and semantic understanding of scene components—such as objects, walls, doors, windows, and rooms—ultimately limiting their effectiveness in real-world deployment. For instance, an agent trained in a kitchen without a refrigerator may

**Instruction**

> In the room, there is a bed, four chairs, a table, and one couch. All walls in the room are ivory. The table is brown. On the table, there are two pencils. **The two pencils are about one meter apart.** The chairs are placed around the side of the table.

**Multi-hop Grounding for Evaluation**

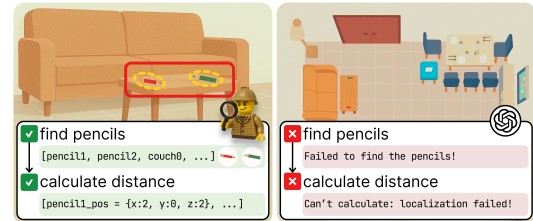

Figure 1: LEGO-EVAL performs multi-hop grounding using tool-retrieved multimodal information (left), whereas VLMs fail to ground pencils in the scene (right).

fail to locate one, open it, and retrieve items during deployment, due to insufficient understanding of the presence and function of the refrigerator. While simulators aim to provide such realism, popular platforms such as AI2THOR (Kolve et al., 2017) rely on manually created 3D environments by experts—a labor-intensive and costly process that severely limits diversity. This limitation in number of scenes can lead to failures in real-world tasks that involve much greater variation in environment.

To scale up the number of scenes, recent work has explored automatic scene generation using Large Language Models (LLMs) and random sampling (Deitke et al., 2022; Yang et al., 2024b). These methods often generate scenes without explicit textual guidance, or rely only on coarse-grained instructions (*e.g.*, "Modern-style kitchen"). While effective for generating many scenes, they frequently yield unrealistic environments—such as a kitchen missing a refrigerator or bookshelves obstructing a window. Training in these scenes can mislead agents to develop incorrect understandings of the physical world. To mitigate this, scene generation can be guided by fine-grained instructions that describe the real-world environment in detail. Thus, fully satisfying the constraints in such detailed instructions is crucial for generating realistic scenes that enable robust agent training.

However, existing evaluation methods cannot reliably verify constraint satisfactions. This unreliability stems from their inability to perform multi-hop grounding: **(1) identifying scene components mentioned in the instruction**, and **(2) verifying their attributes and spatial relationships**. For example, assessing the constraint "a blue chair placed next to the black desk" involves locating both objects in the scene, verifying their colors, and evaluating their spatial relationship. Current approaches lack capability for such multi-hop grounding. One widely adopted method, CLIPScore (Hessel et al., 2021), is inadequate due to the limited capacity of CLIP (Radford et al., 2021) to interpret complex 3D scenes (Hegde et al., 2023; Ma et al., 2023). Similarly, vision-language models (VLMs) used as judges also struggle with precise localization of scene components, as shown in Figure 1, which limits their ability to assess attributes or spatial relations (Li et al., 2024).

To address these limitations, we introduce LEGO-EVAL (**L**anguage-guided **E**nvironment **G**eneration for emb**O**died agents), a comprehensive evaluation framework for assessing text-guided 3D scene synthesis. By utilizing a diverse set of tools, LEGO-EVAL effectively grounds scene components and retrieves relevant information. For rigorous evaluation, our method first identifies constraints within the instruction and then evaluates each individually, providing binary judgments accompanied by detailed, interpretable explanations. We also release LEGO-BENCH, a curated dataset of fine-grained textual instructions that contain constraints about real-world environments. The instructions in the benchmark include diverse attributes and spatial relationships of scene components, capturing a wide range of realistic scene aspects. Together, LEGO-EVAL and LEGO-BENCH provide a robust framework for evaluating text-guided 3D scene synthesis.

Empirical results highlight the effectiveness of our approach. LEGO-EVAL achieves an F1 score of 0.81 and a Cohen's kappa of 0.63 for instruction-scene alignment, demonstrating substantially stronger alignment with human judgments. In contrast, the VLM-as-a-judge baseline shows low agreement, with scores of only 0.40 and 0.05, respectively. We further leverage LEGO-EVAL to benchmark existing LLM-based scene generation methods on LEGO-BENCH, revealing that existing methods achieve success rates of at most 10% in fully satisfying instructions.

## 2 RELATED WORK

**Text-guided 3D scene synthesis.** Early work explored rule-based pipelines, spatial priors, and database composition (Coyne & Sproat, 2001; Chang et al., 2014; Ma et al., 2018). With the advent of neural generative models, diffusion-based pipelines have emerged as a dominant paradigm for text-to-3D synthesis (Höllein et al., 2023; Ma et al., 2024; Tang et al., 2024; Zhou et al., 2025; Fang et al., 2025). Recently, LLMs and VLMs have also been adopted for indoor scene generation, leveraging their real-world priors to act as effective compositional planners. Prior works generally fall into three categories. Scene-level generation approaches (Feng et al., 2023; Yang et al., 2024b; Bucher & Armeni, 2025) generate complete 3D environments directly from language instructions. Object selection and placement methods (Yang et al., 2024a; Çelen et al., 2025) focus on selecting relevant objects and determining their placements. Meanwhile, layout optimization methods (Sun et al., 2025; Ran et al., 2025) aim to optimize the spatial arrangement of predefined assets.

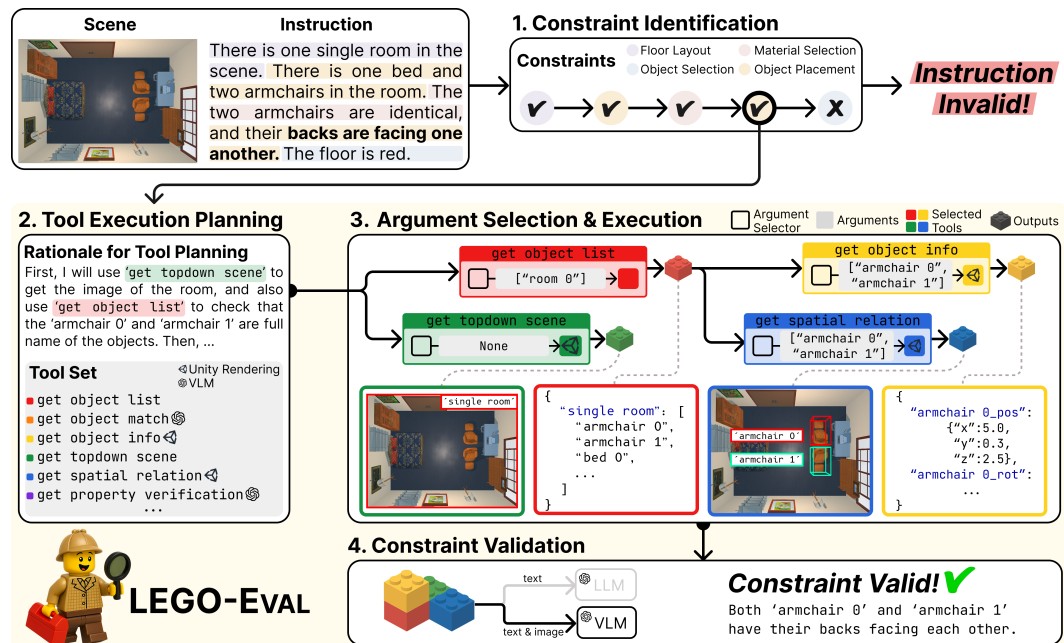

Figure 2: Overview of LEGO-EVAL. LEGO-EVAL plans tool execution using diverse tools, and selects arguments before executing each tool. Constraints are evaluated using the collected outputs.

**Automatic evaluation for 3D scene synthesis.** Evaluating text-guided 3D scene synthesis with fine-grained instructions requires verifying whether generated scene includes all specified constraints, which is a complex task involving multi-hop grounding. For example, assessing "*a blue chair placed next to the black desk*" demands identifying objects, verifying attributes, and checking spatial relationships. Current methods typically rely on two approaches: (1) CLIPScore (Öcal et al., 2024; Fu et al., 2024; Deng et al., 2025), which measures the CLIP similarity between a top-down image of the scene and the instruction, (2) prompting VLMs (Wang et al., 2024b; Çelen et al., 2025; Sun et al., 2025; Ling et al., 2025), providing VLMs with the instruction along with images of the scene captured from multiple viewpoints. However, these methods lack deep understanding of 3D scenes, which limits multi-hop grounding (Ma et al., 2023; Li et al., 2024). To address this, SceneEval (Tam et al., 2025) predefines evaluation criteria such as object count and attributes, and object-object, object-architecture spatial relations, but cannot evaluate attributes and spatial relations of architectures (*e.g.*, "*Sliding window is on orange wall*"). The spatial relations are also predefined to basic relations (*e.g.*, left, near), limiting evaluation of diverse spatial expressions such as "The table is closer to the chair than bed". In contrast, LEGO-EVAL supports spatial reasoning across all scene components and can handle a broad range of relationships expressed in natural language.

**Tool-augmented language models.** LLMs are often augmented with external tools to overcome its limitations of parametric memory. Many works augment LLMs with external tools to improve the aspects such as factuality (Komeili et al., 2022), math abilities (Imani et al., 2023), and solve complex tasks (Paranjape et al., 2023; Chen et al., 2023; Patil et al., 2024). There has also been growing interest in augmenting language models with multimodal tool use. VisProg (Gupta & Kembhavi, 2023) and ViperGPT (Surís et al., 2023) integrate vision modules with Python-based text utilities to decompose and solve visual reasoning tasks. In contrast, AVIS (Hu et al., 2023) and Chameleon (Lu et al., 2023) combine vision tools with textual functions such as knowledge retrieval or tabular processing, targeting broader multimodal information-seeking and compositional reasoning. Our evaluation method similarly augments VLMs with multimodal tools for environment interaction, textual reasoning, and multimodal reasoning, supporting multi-hop grounding in 3D scenes.

## 3 LEGO-EVAL: EVALUATION WITH TOOL-AUGMENTED VLMS

Evaluating 3D scenes requires localizing individual scene components, and retrieving their detailed information about attributes and spatial layout. To support this, LEGO-EVAL is augmented with a diverse suite of tools capable of retrieving both visual and textual information from the scene.

### 3.1 EVALUATION FRAMEWORK

Leveraging diverse tools, our framework aims to rigorously evaluate the alignment between fine-grained instructions and generated 3D scenes. Specifically, given fine-grained instruction $I$ and generated scene $S$, evaluator provides binary judgment $J$ along with evaluation explanations $E$:

$$J, E \leftarrow \text{Eval}(I \mid S) \tag{1}$$

As shown in Figure 2, our evaluation framework consists of the following four steps:

**Step 1: Constraint Identification.** To enable rigorous evaluation, we begin by identifying constraints $C = (c_1, \ldots, c_k)$ within each instruction $I$. Fine-grained instructions include multiple constraints, each contributing to aspects of the desired scene. LEGO-EVAL identifies these constraints and categorizes them into one of four types, similar to the modules in Holodeck (Yang et al., 2024b):

- **Floor Layout**: Constraints that define the spatial layout of rooms, walls, doors, and windows.
- **Material Selection**: Constraints specifying the visual appearance of floors and walls.
- **Object Selection**: Constraints describing the appearance of objects, including doors and windows.
- **Object Placement**: Constraints that determine object placement and rotation within the scene.

**Step 2: Tool Execution Planning.** The constraints identified in the previous step are evaluated sequentially. Given the current constraint and explanations of prior constraint evaluations, the model generates a tool execution plan along with the rationale behind its planning decisions. Often, the necessary information for evaluation has already been retrieved from earlier assessments. For instance, verifying whether "*the cup is on the red table*" first requires confirming the existence of a red table, which may have been already confirmed in earlier constraint evaluations. Therefore, by leveraging such prior results, the model avoids redundant tool executions. Based on the constraint and prior evaluations, LEGO-

| Tool Types | Tools |
|---|---|
| ◁ **Environment Interaction** | get topdown scene, get topdown room, get frontview object, get wall scene, get topdown object, get material image get multiview rendered object, get spatial relation |
| ▦ **Textual Reasoning** | get room list, get room info, get wall list, get wall info, get door list, get door info, get window list, get window info, get object list, get object info |
| ▦ **Multimodal Reasoning** | get object match, get property description, get property verification |

Figure 3: Diverse tools included in our tool set.

EVAL selects tools from the tool set that excludes those irrelevant to the constraint type. Then, it generates a graph-structured execution plan that supports parallel tool executions, enabling efficient evaluation.

**Step 3: Tool Argument Selection & Execution.** Accurate validation depends not only on selecting the right tool but also on choosing appropriate arguments. Without appropriate arguments, even the correct tool may return irrelevant output. To guide this process, we provide LEGO-EVAL —prior to each tool execution—with contextual input that helps the model infer the tool's intended role and determine the information needed for validation. Specifically, the instruction, constraint, execution plan, and its rationale enable the model to infer the tool's intended function within the current context and clarify what information needs to be extracted. The model then selects appropriate arguments by drawing on both prior tool outputs (*e.g.*, a list of rooms) and explanations from earlier constraint evaluations (*e.g.*, "The table (id: table-2) is red"), which identify scene components in context. Once the arguments are selected, the tool is executed to retrieve the information of those components.

**Step 4: Constraint Validation.** After executing all tools, the model assesses whether the generated scene satisfies each constraint based on the corresponding tool outputs. A scene is deemed valid only if it fulfills all constraints $C$ specified in the instruction $I$.

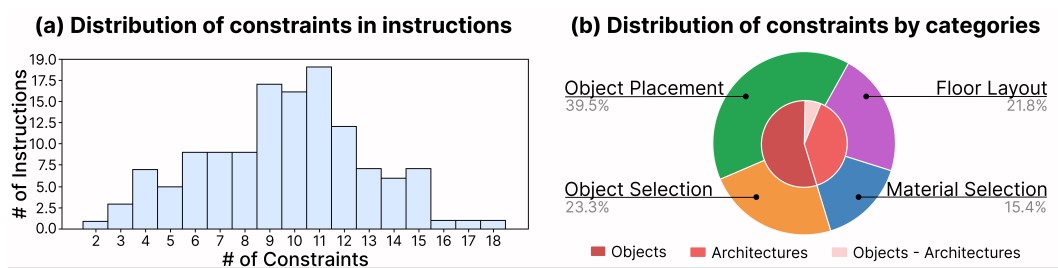

Figure 4: Statistics of LEGO-BENCH.

## 3.2 TOOL SET

To robustly ground scene components and accurately retrieve their attributes and spatial configurations, we enhance LEGO-EVAL with 21 diverse tools (Figure 3). These tools are grouped into three distinct types, each designed to address a specific aspect of scene understanding:

- **Environment Interaction**: These tools interact with the Unity environment to retrieve visual information such as object appearance and spatial arrangements, that text cannot fully represent.
- **Textual Reasoning**: These tools retrieve textual descriptions from structured scene representations such as exact coordinates or occluded object attributes, that image cannot reliably provide.
- **Multimodal Reasoning**: These tools convert visual information into textual descriptions to retrieve specific information from images. Since VLMs struggle with multi-image inputs (Wang et al., 2024a), we use LLMs and VLMs to convert images into text descriptions.

Descriptions of each tool can be found in Appendix C.3.

## 3.3 LEGO-BENCH

Real-world indoor environments are composed not only of objects but also of architectural components such as walls, doors, windows, floors, and room layouts. Reflecting this, we introduce LEGO-BENCH —a benchmark designed to evaluate LLM-based 3D scene synthesis methods, with a focus on both the attributes and spatial relationships of all scene components.

**Dataset collection.** We manually collect instructions for 3D scene synthesis. These instructions describe scenes with multiple constraints, curated based on real-world images and empirical observations of indoor spaces. To enable fair comparison between methods, each instruction is annotated with constraints and constraint types. We also provide manually curated scenes that fully satisfy the instructions. Further details on our dataset collection procedure can be found in Appendix B.2.

**Statistics.** Figure 4 summarizes key statistics of our collected dataset. The LEGO-BENCH benchmark includes 130 natural- language instructions paired with manually annotated scenes, encompassing a total of 1,250 constraints. On average, each instruction specifies 9.6 constraints, with the majority falling between 9 and 11. These constraints span a broad range of scene elements: 55% involve objects, while 39% target architectural components. We further categorize constraints based on their semantics—approximately 40% relate to material and object selection, and the remaining 60% involve floor layout and object placement. Together, these distributions highlight the complexity and diversity of real-world environments captured by LEGO-BENCH.

## 4 EXPERIMENTS

To evaluate the effectiveness of LEGO-EVAL, we conduct experiments in evaluating text-guided 3D scene synthesis, focusing on the agreement with human judgments. Additionally, existing LLM-based 3D scene synthesis methods are benchmarked on LEGO-BENCH, revealing their limitations.

| Methods | Holistic ↑ | | | | Partial ↑ | | | |
|---|---|---|---|---|---|---|---|---|
| | F1 | Recall | Precision | Cohen's $\kappa$ | F1 | Recall | Precision | Cohen's $\kappa$ |
| **SceneEval*** | | | | | | | | |
| Full Dataset | 0.33 | 0.50 | 0.25 | 0.00 | 0.28 | 0.43 | 0.39 | 0.00 |
| Measurable Dataset | 0.47 | 0.58 | 0.74 | 0.15 | 0.45 | 0.58 | 0.64 | 0.12 |
| **CLIPScore** | | | | | | | | |
| Threshold=15 | 0.37 | 0.52 | 0.67 | 0.03 | 0.43 | 0.50 | 0.73 | 0.01 |
| Threshold=20 | 0.49 | 0.51 | 0.51 | 0.02 | 0.46 | 0.50 | 0.50 | 0.00 |
| Threshold=25 | 0.42 | 0.50 | 0.51 | 0.01 | 0.51 | 0.55 | 0.54 | 0.07 |
| **VLM-as-a-Judge** | | | | | | | | |
| Gemini 2.5 Pro | 0.38 | 0.52 | 0.76 | 0.05 | 0.60 | 0.70 | 0.66 | 0.28 |
| GPT-o4-mini | 0.40 | 0.53 | 0.70 | 0.05 | 0.67 | 0.75 | 0.70 | 0.39 |
| GPT-4.1 | 0.40 | 0.53 | 0.67 | 0.05 | 0.68 | 0.73 | 0.65 | 0.35 |
| **Ours** | | | | | | | | |
| GPT-4.1 | **0.81** | **0.82** | **0.84** | **0.63** | **0.83** | **0.81** | **0.86** | **0.66** |
| GPT-4.1-mini | 0.70 | 0.72 | 0.78 | 0.43 | 0.78 | 0.76 | 0.80 | 0.56 |
| Qwen2.5VL-32B | 0.64 | 0.66 | 0.70 | 0.32 | 0.72 | 0.71 | 0.73 | 0.44 |

Table 1: Comparison of evaluation methods. * denotes method that cannot evaluate all constraints; it is assessed on evaluable subset (Measurable) or by treating unevaluable as incorrect (Full).

## 4.1 COMPARISON OF EVALUATION METHODS

### 4.1.1 SETUP

**Dataset.** We use the instructions, annotated constraints, and scenes from LEGO-BENCH to compare the performance of evaluation methods, relying on a fixed set of constraints to ensure fair comparison. To enrich the dataset, we also manually curate 130 additional scenes that intentionally do not fully satisfy the instructions. This results in a total of 260 instruction-scene pairs.

**Baselines.** We compare our performance with SceneEval (Tam et al., 2025), CLIPScore (Hessel et al., 2021), and VLM-as-a-judge. CLIPScore performs binary judgment on the instruction and top-down scene image using thresholds of 15, 20, and 25. For VLM-as-a-judge, we provide scene images from four perspectives and use Gemini-2.5-Pro, GPT-4o mini, and GPT-4.1 as base models with self-consistency across 3 samples for fair comparison. Following LEGO-EVAL, all baseline methods consider a scene aligned with the instruction if all constraints in the instruction are satisfied.

**Metrics.** We assess the performance of evaluation methods using F1 score, precision, recall, and Cohen's kappa. While F1 captures the balance between precision and recall, Cohen's kappa measures agreement with human judgments beyond chance, offering a more reliable and robust view of overall evaluation quality. To support both broad and detailed analysis, we compute these metrics at two distinct levels: (1) Holistic: measures the agreement with human judgments on the full instruction, and (2) Partial: measures the agreement with human judgments on each individual constraint.

### 4.1.2 RESULTS

In Table 1, we present a comparative evaluation of LEGO-EVAL against existing methods. SceneEval (Tam et al., 2025), constrained by a fixed set of metrics and criteria, is unable to assess 41% of the constraints in the dataset. To enable a fair comparison, we evaluate its performance under two settings: treating unevaluable constraints as incorrect, and modifying instructions to include only those constraints it can process. SceneEval also suffers from a fundamental limitation—it fails to reliably ground scene components referenced in the instructions. While accurate evaluation depends first on identifying the relevant objects, SceneEval often fails at this initial step.

Predominantly used methods exhibit similar limitations. CLIPScore (Hessel et al., 2021) is unreliable for fine-grained alignment with 3D scenes, as CLIP (Radford et al., 2021) is trained on 2D image–text pairs and lacks 3D spatial understanding. VLM-as-a-judge also frequently misidentifies or fails to localize mentioned components, resulting in incorrect constraint assessments. In contrast, LEGO-EVAL achieves higher F1 and Cohen's kappa scores at both the constraint and instruction levels. By integrating diverse tools for multi-hop grounding, LEGO-EVAL more effectively locates and interprets scene components—enabling accurate and robust evaluation of 3D scene synthesis.

Figure 5: Distribution of tool types executed by LEGO-EVAL during evaluation.

| Method | Holistic SR ↑ | Partial SR ↑ | | | | |
|--------|---------------|--------------|--------------------|------------------|------------------|------|
| | | Floor Layout | Material Selection | Object Selection | Object Placement | Avg. |
| I-Design | 3.8 | 92.7 | 63.7 | 11.0 | 4.1 | 34.2 |
| LayoutGPT | 6.9 | 96.0 | **65.3** | 40.9 | 37.3 | 55.2 |
| Holodeck | 8.4 | **96.3** | 61.6 | 46.3 | 43.7 | 58.5 |
| LayoutVLM | **10.0** | 95.6 | **65.3** | **49.8** | **46.0** | **60.6** |

Table 3: Evaluation results of LLM-based 3D scene synthesis methods on LEGO-BENCH.

### 4.1.3 ABLATION STUDY

We investigate the impact of disabling specific tool types and analyze the resulting performance of LEGO-EVAL. Since tools returning list of scene components are necessary for argument selection, these remain enabled. While conventional evaluation methods often rely solely on visual inputs, the results in Table 2 demonstrate that textual information is also critical for rigorous assessment. For instance, text can capture subtle scene attributes—such as the color of small objects—that are difficult to infer from images alone, and it can also convey key information about

| Tools | Holistic F1 (Δ) | Partial F1 (Δ) |
|-------|-----------------|----------------|
| w/o **M** ⊞ | −0.04% | −1.02% |
| w/o **T** 🔲 | −5.05% | −2.65% |
| w/o **T** 🔲 + **M** ⊞ | −6.46% | −2.81% |
| w/o **E** ◁ + **M** ⊞ | −24.90% | −5.34% |

Table 2: Performance drop with tool types disabled. (**M**: Multimodal Reasoning, **T**: Textual Reasoning, **E**: Environment Interaction)

scene components extracted from retrieved images. Figure 5 further supports this finding by showing that all tool types are actively used across different constraint types. These observations collectively highlight that all three tools are indispensable for comprehensive and reliable evaluation.

### 4.2 COMPARISON OF TEXT-GUIDED 3D SCENE SYNTHESIS METHODS

#### 4.2.1 SETUP

**Baselines.** We evaluate four LLM-based 3D scene synthesis methods on LEGO-BENCH: Layout-GPT (Feng et al., 2023), Holodeck (Yang et al., 2024b), I-Design (Çelen et al., 2025), and LayoutVLM (Sun et al., 2025). These methods vary in functionality: Holodeck generates complete scenes with object selection, attributes, and placement; I-Design selects and places objects; Layout-GPT and LayoutVLM position a given set of objects. To enable fair comparison, we augment the latter three with Holodeck to produce full scenes for evaluation.

**Metrics.** For evaluation, we use the following Success Rates (SR): (1) Holistic SR: The proportion of instructions with all specified constraints satisfied. (2) Partial SR: The proportion of valid constraints. For partial SR, we report success rates by constraint type and their overall average.

#### 4.2.2 RESULTS

**Main results.** We benchmark LLM-based 3D scene synthesis methods on LEGO-BENCH to provide a comprehensive assessment of existing approaches. As shown in Table 3, most methods achieve average partial SRs exceeding 50%, yet their performance on object selection and placement is lower. This indicates that current methods struggle to effectively handle these fundamental aspects of scene synthesis. Interestingly, while selecting appropriate objects affects the performance of object placement, we also observe vice versa. Although LayoutGPT (Feng et al., 2023) and LayoutVLM use Holodeck (Yang et al., 2024b) for object selection, their performance differs from that of Holodeck. This stems from differences in how these methods position the selected objects, with some methods failing to place them in the scene.

| Methods | Holistic SR | | | | Partial SR | | | |
|---|---|---|---|---|---|---|---|---|
| | M1 | M2 | M3 | M4 | M1 | M2 | M3 | M4 |
| Oracle Constraint | 0.12 | 0.05 | 0.05 | 0.14 | 0.66 | 0.57 | 0.35 | 0.65 |
| Identified Constraint | 0.13 | 0.07 | 0.05 | 0.12 | 0.63 | 0.57 | 0.35 | 0.64 |
| Difference in SR | +0.01 | +0.02 | +0.00 | -0.02 | -0.03 | +0.00 | +0.00 | -0.01 |

Table 4: End-to-end evaluation maintains consistent results with annotated constraint evaluation.

| Models | Component Performance | | | Evaluation Performance | |
|---|---|---|---|---|---|
| | Tool F1 ↑ | GED ↓ | Argument F1 ↑ | Holistic F1 ↑ | Partial F1 ↑ |
| Gemma3-27B | 0.50 | 3.01 | 0.38 | 0.61 | 0.68 |
| Qwen2.5VL-32B | 0.57 | 2.87 | 0.46 | 0.64 | 0.72 |
| Qwen3-32B | 0.62 | 2.55 | 0.42 | 0.69 | 0.73 |

Table 5: The performance of components are highly correlated with the evaluation performance. Tool F1 and argument F1 measure prediction accuracy; GED measures graph structural similarity.

**Satisfying all constraints in an instruction is challenging.** The substantial disparity between partial and holistic success rates (SR) across all baseline methods in Table 3 underscores a critical limitation: current approaches struggle to satisfy all constraints within a single instruction. To analyze this further, we categorize instructions by constraint complexity into three groups: simple (2–7 constraints), moderate (8–12), and complex (more than 12). We also collect user-generated descriptions of individual rooms, reflecting realistic design scenarios (see Appendix D.2). Notably, these descriptions contain an average of 18.2 constraints per room. As shown in Figure 6, existing methods consistently fail on complex instructions, revealing their inability to generate scenes that fully satisfy the constraints commonly found in real-world environments.

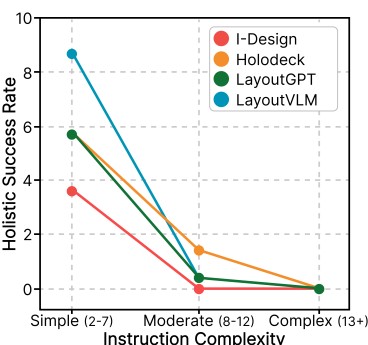

Figure 6: Holistic SR drops as the complexity of the instruction rises.

## 5 ANALYSIS

**LEGO-EVAL provides end-to-end automated evaluation.** To assess the effectiveness of automated constraint extraction, we compare LEGO-EVAL 's performance using its own automatically identified constraints against its performance with human-annotated ones. Accurate, fine-grained evaluation requires capturing all relevant constraints in a given instruction. In our setup, constraints are extracted and categorized using GPT-4.1, then used to evaluate 3D scene generation outputs from Qwen2.5VL-32B. These evaluations are directly compared with those conducted using human-annotated constraints. As shown in Table 4, LEGO-EVAL exhibits only minor performance differences across four 3D scene synthesis methods when using automatic versus human-provided constraints. This close alignment demonstrates that LEGO-EVAL can reliably identify and classify constraints, making it suitable as a fully automated, end-to-end evaluation tool.

**Effective evaluation depends on appropriate tool planning.** We investigate the relationship between evaluation performance and two key components of the evaluation process: tool execution planning and argument selection. We evaluate both components on all constraints introduced in Section 4.1. For tool planning, LEGO-EVAL generates execution plans conditioned on the given constraints. For argument selection, LEGO-EVAL selects arguments given a human-annotated tool execution plan. We test multiple LLMs for components to observe their contributions, using Qwen2.5VL-32B as a fixed validator. We use three metrics to assess component performance: (1) Tool F1, the macro F1 score over constraints, where a tool prediction is correct if it appears in the ground-truth set; (2) Argument F1, the macro F1 over tools, with correctness based on ground-truth arguments; and (3) Graph Edit Distance (GED), measuring the number of edits needed to align predicted and ground-truth constraint graphs. Evaluation performance is computed using multiple

**Instruction**

There is only one room in the scene. The room is square shaped. There are two large windows on one side of the room. There is one laptop, one flashlight, and two chairs. The flashlight is cylindrical, and has a metallic head. **The flashlight and the laptop are facing the same direction.** There is also a plant, which is planted in a pot that has red ribbon around it.

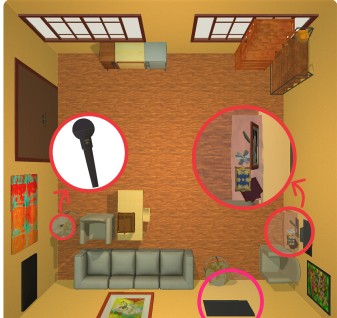

**VLM-as-a-Judge**    *Invalid* ✗

The laptop is on a side table under the TV on the right wall. The flashlight lies on the round table next to the armchair. **Since the laptop faces into the room and the flashlight faces toward the wall, their orientations do not match.**

**SceneEval**    *Invalid* ✗

**There is a laptop in the scene. There is no flashlight in the scene.** Not all objects in the constraint are present in the scene. Thus, the constraint is not satisfied.

**LEGO-Eval**    *Valid* ✔

**Since neither object is present, there is no way to assess whether the flashlight and the laptop are facing the same way.** This means the constraint cannot be satisfied.

Figure 8: Comparison of evaluation results from VLM-as-a-judge, SceneEval, and LEGO-EVAL.

LLMs, with Qwen2.5VL-32B as the validator. Results in Table 5 show that tool execution planning correlates more strongly with overall evaluation performance than argument selection, highlighting its role as an orchestrator. When tool plans are fixed to ground truth, argument selection correlates more with performance, emphasizing its dependence on accurate planning (See Section D.1). This suggests effective tool planning is critical for appropriate argument selection and evaluation.

**LEGO-EVAL as a feedback signal for refinement.** To illustrate the reliability and interpretability of evaluations, we refine the results of Holodeck (Yang et al., 2024b) from Section 4.2 with the feedback from LEGO-EVAL. Specifically, we use LEGO-EVAL to evaluate scenes generated by Holodeck, then provide feedback to refine invalid scenes. We iterate this refinement process 3 times, comparing LEGO-EVAL against prompted VLMs as feedback signals. The results show that Holodeck achieves higher performance using LEGO-EVAL as feedback than prompted VLMs, demonstrating LEGO-EVAL's superior feedback quality for refinement. A possible explanation is that while prompted VLMs often follow invalid reasoning paths, LEGO-EVAL uses valid reasoning, which is reflected in its detailed, interpretable evaluation explanations.

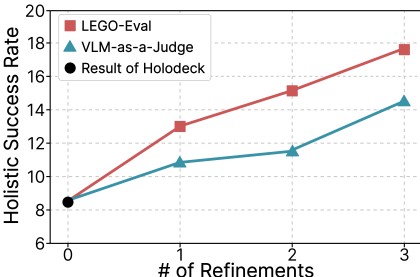

Figure 7: Comparison of refinement results using VLM-as-a-judge and LEGO-EVAL as feedback signal.

**Case Study.** Figure 8 illustrates an example comparing evaluation results from VLM-as-a-judge, SceneEval, and LEGO-EVAL, revealing that while all methods achieve accurate judgments, their reasoning processes differ significantly. Although the flashlight and laptop do not exist in the scene, the VLM-as-a-judge locates them and determines they are not facing the same direction. Similarly, SceneEval misidentifies the black painting on the wall as a laptop. In contrast, LEGO-EVAL accurately recognizes the absence of both objects and determines the constraint cannot be satisfied.

# 6 CONCLUSION

In this work, we introduce LEGO-EVAL, a comprehensive evaluation framework for text-guided 3D scene synthesis, along with a robust benchmark comprising fine-grained instructions with multiple constraints designed to reflect real-world complexity. Our experimental results show that our approach more than doubles the F1 score compared to the baseline, demonstrating significant improvements in robustness. We also reveal that existing scene generation methods achieve a success rate of only 10%, underscoring the current limitations of LLM-based approaches in 3D scene generation. We believe this framework will support progress toward generating scenes that faithfully reflect real-world specifications, ultimately enabling more capable and reliable embodied agents.

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

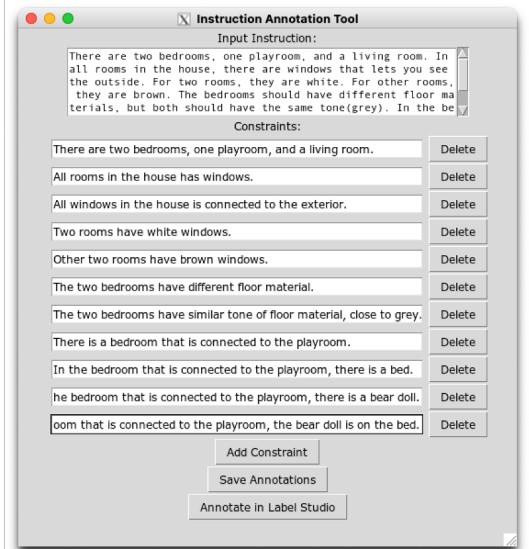

Figure 9: Our annotation tool used for annotating constraints.

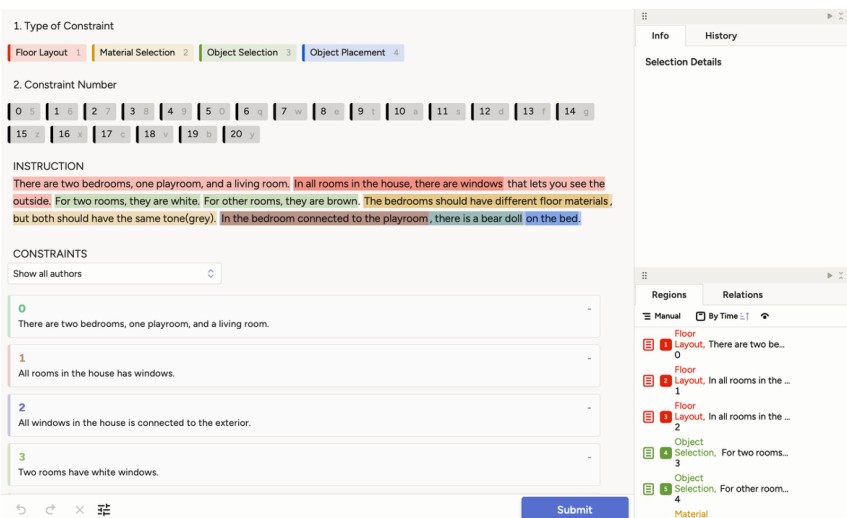

Figure 10: Interface of Label Studio for constraint classification.

## A THE USAGE OF LARGE LANGUAGE MODELS (LLMs)

This work made limited use of Large Language Models (LLMs), primarily as an aid for refining the writing of this paper (e.g., grammar and expression). LLMs were also used to provide partial assistance in writing experimental code. All research design decisions, methodological development, dataset curation, and the writing of the paper's substantive content were conducted and verified by the authors, who take full responsibility for the contents presented.

## B LEGO-BENCH

### B.1 DATA ANNOTATION TOOL

To reduce burden of human annotation, we developed a tool for constraint identification, and use Label Stuio for constraint classification, Figure 9 shows the interface of the tool for annotating the constraints within the instructions. This tool allows the annotators to track if all constraints are ac-

curately identified. Figure 10 shows the interface of Label Studio for annotating the constraint types and mapping each constraint with the corresponding parts of the instruction. The constraint type is for annotating the constraints to the correct type, and constraint number is to map the constraint to the appropriate part of the instruction.

## B.2 DETAILS OF HUMAN ANNOTATION

Our data collection process follows the process below:

**Step 1: Annotator recruiting and education.** To collect data, we recruit human annotators to construct the dataset. All annotators completed a two-hour education session prior to annotation. This education covers a detailed explanation about guidelines for writing quality instructions and identifying constraints, data annotation interface, and generating scenes that align with the instruction. After completing the training, each annotator is assigned to create instructions, annotate them, and generate corresponding aligned scenes.

**Step 2: Data annotation.** The annotation process is structured into three phases. In the first phase, we ask each annotator to create 30 to 50 fine-grained instructions describing the scenes. Annotators are instructed to create instructions that reflect real-world scenes relying on their empirical experiences.

In the second phase, annotators annotate each instruction by identifying the constraints included in the instruction using our annotation tool 9 and Label Studio 10. As some constraints can only be satisfied when others are fulfilled, we express these dependencies explicitly. We achieve this by restructuring the constraints into conditional expressions that directly reference their prerequisites. For example, "There is a room with a red floor. There is a bed in that room. The bed is white." The sentence "The bed is white" relies on the preceding context, as it refers to the bed in the room with the red floor. This constraint can then be expressed as "*The bed in the room with the red floor is white.*" We also merge multiple attributes of the same entity into unified statements, such as combining "the three walls are yellow" and "the three walls have patterns" into "*the three walls are yellow and have patterns.*" Once the constraints have been revised, each constraint is annotated with one of four constraint types: floor layout, material selection, object selection, or object placement. It is also labeled with its corresponding text segment in the instruction. After identifying the constraints, the annotators label each constraint with appropriate constraint type. We also annotate which parts of each instruction correspond to specific constraints, similar to entity annotation in Named Entity Recognition (NER).

In the third phase, annotators create a scene that fully validates the constraints within the instruction. To achieve this, we first input the instructions into Holodeck (Yang et al., 2024b), which generates scenes in textual representations (*i.e.*, JSON scripts) that can be rendered in the 3D simulator. The textual representations contain comprehensive scene information, including the asset ID of each component as well as the exact position and rotation of all scene components. As shown in Table 4.2, Holodeck cannot generate scenes that fully align with the instructions. Thus, annotators manually modify the text representation of the scene so that the scene fully aligns with the instruction.

**Step 3: Verification.** To ensure the quality of LEGO-BENCH, we conduct manual verification. Specifically, annotators conduct mutual reviews of the annotated instructions and corresponding scenes. Annotators first verify whether the constraints have been correctly identified and whether the annotations comply with the instruction annotation guidelines. The instructions are re-annotated if the constraints are not accurately identified. Annotators also verify whether the generated scene fully aligns with the instruction. If misalignment is found, the annotators modify the scene to ensure the alignment. The procedure is repeated for two iterations.

## B.3 STATISTICS

The overall statistics of the LEGO-BENCH are shown in Table 6. To illustrate the diversity of our collected instructions, we utilize a word cloud representation in Figure 11. The visualization highlights the diverse linguistic forms and semantic categories present in the LEGO-BENCH, indicating that the instructions span across multiple domains and levels of abstraction.

| Category | Value |
|---|---|
| # Instructions | 130 |
| Avg. # Rooms per Instruction | 2.27 |
| Avg. Instruction Length (words) | 98.54 |

Table 6: Statistics of LEGO-BENCH.

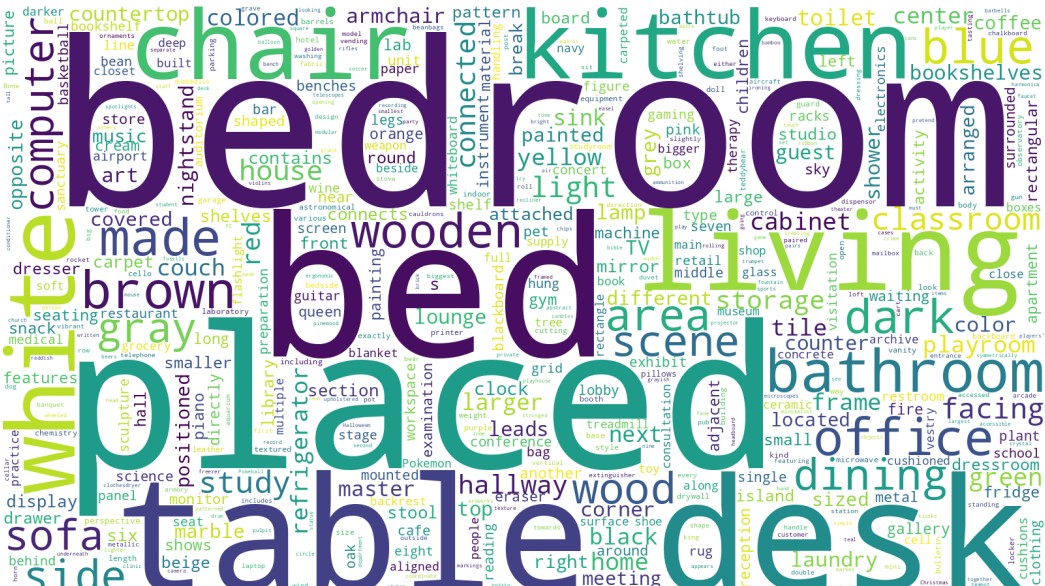

Figure 11: Word cloud illustrating the diversity of instructions in LEGO-BENCH.

## C    LEGO-EVAL

We provide additional implementation details of LEGO-EVAL in the following.

### C.1    MODEL INFERENCE

We prompt the closed-source models, including Gemini 2.5 Pro, o4-mini, gpt-4.1-2025-04-14, and gpt-4.1-mini-2025-04-14, for both LLM and VLM inference. The input images are resized to 1200px resolution, and the temperature is fixed at 0 to ensure deterministic outputs.

For the open-source model Gemma3-27B, Qwen2.5VL-32B, and Qwen3-32B, we use the default hyperparameter settings provided by the model for inference. The input images are resized to 335px resolution; when the number of images exceeds the model's processing capacity, two images are concatenated into a single input, with their corresponding names displayed at the top-left corner of each image.

### C.2    METRIC DETAILS

To aid readers' understanding, we provide detailed explanations of the evaluation metrics below.

- **F1 Score**: This metric evaluates the balance between precision and recall, serving as the harmonic mean of the two. A higher F1 score indicates better performance in correctly identifying positive instances while avoiding false positives and false negatives.

$$F1 = \frac{2 \cdot \text{Precision} \cdot \text{Recall}}{\text{Precision} + \text{Recall}} \tag{2}$$

- **Cohen's Kappa** ($\kappa$): This metric measures the level of agreement between two raters, adjusting for the amount of agreement that could occur by chance. A higher $\kappa$ value indicates

stronger agreement, where $\kappa = 1$ denotes perfect agreement and $\kappa = 0$ corresponds to chance-level agreement. We use Cohen's kappa to measure the agreement with human judgments.

$$\kappa = \frac{p_o - p_e}{1 - p_e} \tag{3}$$

## C.3 TOOL SET

In this section, detailed descriptions for each tool are provided. For tools that output images, example outputs of the images for each tool are also provided.

### C.3.1 ENVIRONMENT INTERACTION

- **get topdown scene:** This tool generates a bird's-eye (top-down) view of the environment. It produces a dictionary containing an image array that represents the entire scene layout, including walls, doors, and objects. The example output is shown in Figure 15.

- **get topdown room:** This tool generates a top-down visual representation of a specific room in the environment. By providing the room ID as input, it produces a structured dictionary containing an image array that depicts the layout of the room, including walls, doors, and objects. The example output is shown in Figure 16.

- **get frontview object:** This tool produces front-view images of specified objects within a scene. Given a list of object IDs, it centers each target object in the frame while preserving surrounding context, such as nearby objects, for richer spatial awareness. The output is a structured dictionary mapping each object ID to its corresponding front-view image array. The example output is shown in Figure 17.

- **get wall scene:** This tool generates clear wall-view images of a room from specified directions. By taking a list of wall IDs as input, it produces perspective images where objects in the middle of the room are removed to prevent occlusion, ensuring unobstructed visibility of the walls. The output is a dictionary mapping each wall to its corresponding image array. The example output is shown in Figure 18.

- **get topdown object:** This tool produces object-centric images from an overhead perspective. Given a list of object IDs, it generates top-down views where each target object is centered in the frame, while preserving surrounding context such as nearby objects. The resulting dictionary maps each object to image arrays, allowing inspection at different distances. The example output is shown in Figure 19.

- **get material image:** This tool generates visual representations of specified materials, restricted to walls and floors within the scene. By providing a list of material names, it returns a dictionary mapping each material to an RGBA image array. The tool only includes materials that are actually present in the designated environment. The example output is shown in Figure 20.

- **get multiview rendered object:** This tool produces rendered images of specified objects using their underlying asset IDs. Given a list of object IDs, it returns a dictionary where each object ID maps to an image array showing the rendered appearance of that object. The example output is shown in Figure 21.

- **get spatial relation:** This tool generates scene images that isolate and display only specified object pairs (or sets), such as objects, windows, doors, or walls. By taking a list of tuples containing object IDs, it produces a dictionary mapping each group of objects to an image array. Values of the dictionary provide visualizations that highlight the spatial relationships among the selected elements. The example output is shown in Figure 22.

### C.3.2 MULTIMODAL REASONING

- **get object match:** This tool provides semantic descriptions for object IDs by mapping them to their corresponding object types. It is designed to be used after retrieving rendered views with `get multiview rendered object`, ensuring accurate alignment between visual assets and their categorical labels. The output is a dictionary that links each object ID to a descriptive type.

- **get property description:** This tool generates semantic descriptions of objects by identifying their color, shape, and material from rendered images and optional textual metadata. It is intended for use after retrieving visual inputs with `get material image` and `get multiview rendered object`, ensuring accurate grounding of appearance-based attributes. The output is a dictionary summarizing the derived text information, along with reasoning.

- **get property verification:** This tool validates descriptive attributes of a subject—either an object or material—based on a given instruction. First, an LLM parses the instruction to determine which attributes should be checked (e.g., surface patterns or textures). Then, a VLM analyzes the corresponding images to extract those attributes. It is designed for use after obtaining visuals with `get material image` and `get multiview rendered object`. The output is a dictionary summarizing the verified textual information.

### C.3.3 Textual Reasoning

- **get room list:** This tool extracts all room identifiers available within a given scene. It returns a dictionary which contains a list of room IDs as strings.

- **get room info:** This tool provides detailed metadata for specific rooms in a scene. Given a dictionary from `get room list` containing room IDs, it returns structured information about each room. The output includes the polygon vertices defining the room's geometry and the identified floor material.

- **get wall list:** This tool extracts the wall identifiers associated with specific rooms in a scene. Given a list of room IDs, it returns a dictionary mapping each room ID to a list of its connected wall IDs.

- **get wall info:** This tool provides comprehensive metadata for specified walls in a scene. Using a list of wall IDs (retrieved via `get wall list`), it returns the values including details such as the wall's unique ID, connected room IDs, geometric coordinates, material type, width, height, and directional orientation.

- **get door list:** This tool extracts the door identifiers connected to specific rooms within a scene. Given a list of room IDs, it returns a dictionary mapping each room ID to its associated door IDs.

- **get door info:** This tool provides detailed metadata for doors within a scene. Using a list of door IDs (obtained via `get door list`), it returns the values capture attributes such as the door's unique ID, associated asset ID, connected rooms, adjoining walls, 3D position, and openness state.

- **get window list:** This tool extracts the window identifiers associated with specific rooms in a scene. Given a list of room IDs, it returns a dictionary mapping each room ID to its corresponding window IDs.

- **get window info:** This tool provides detailed metadata for windows in a scene. Given a list of window IDs (retrieved via `get window list`), it outputs the values include attributes such as the window's unique ID, associated asset ID, connected rooms, adjoining walls, and 3D position.

- **get object list:** This tool extracts the identifiers of objects contained within specific rooms of a scene. Given a list of room IDs, it returns a dictionary mapping each room ID to the corresponding list of object IDs present in that room.

- **get object info:** This tool provides detailed metadata about objects in a scene. Using a list of object IDs (retrieved via `get object list`), it returns the values include attributes such as the object's unique ID, associated asset ID, room assignment, 3D position, rotation, and geometric representation through coordinates or mesh vertices.

| Models | Argument Selection | | |
|---|---|---|---|
| | Argument F1 ↑ | Holistic F1 ↑ | Partial F1 ↑ |
| Gemma3-27B | 0.38 | 0.65 | 0.66 |
| Qwen2.5VL-32B | 0.46 | 0.68 | 0.68 |
| Qwen3-32B | 0.42 | 0.68 | 0.67 |

Table 7: Evaluation performance of models on argument selection.

# D ADDITIONAL RESULTS

## D.1 ARGUMENT SELECTION HAS THE AFFECTS ON THE EVALUATION PERFORMANCE.

To further analyze the contribution of argument selection in our evaluation pipeline, we conduct an exper where the tool sequence is fixed to human-annotated ground truth, and models are only responsible for selecting appropriate arguments for each tool. All other components of LEGO-EVAL, including tool execution and validation, use Qwen2.5VL-32B as the backbone.

We evaluate three open-source models Gemma3-27B, Qwen2.5VL-32B, and Qwen3-32B on their ability to perform argument selection. As shown in Table D.1, Qwen2.5VL-32B achieves the highest Argument F1 score, aligning with superior holistic and partial evaluation performance. Qwen3-32B also performs competitively, while Gemma3-27B lags behind across all metrics.

These results highlight that accurate argument selection is crucial for reliable evaluation. Since argument identification directly determines the quality of evidence retrieved by tools, improvements in this component significantly enhance both holistic and partial agreement with human judgments.

## D.2 DO PEOPLE REALLY GIVE LONG INSTRUCTIONS TO DESCRIBE THE ROOM?

To answer this question, we verify whether people actually provide instructions to an LLM for room generation that are as long as we claimed. Therefore, as shown in Figure 12, we conduct a survey where participants are asked to describe a room photo in the form of a prompt. The results show that, on average, the instructions contain 18.2 constraints. This finding confirms that people indeed provide complex instructions, and highlights the importance of LEGO-EVAL, which can robustly evaluate generated rooms based on such instructions.

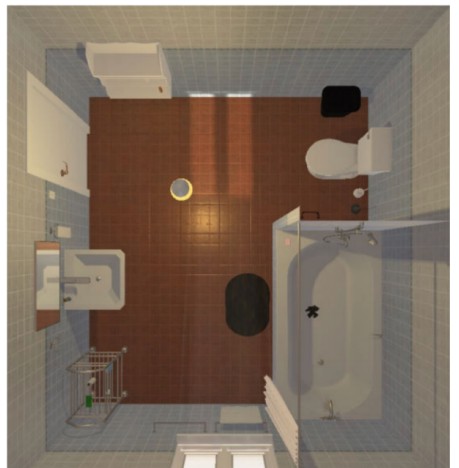
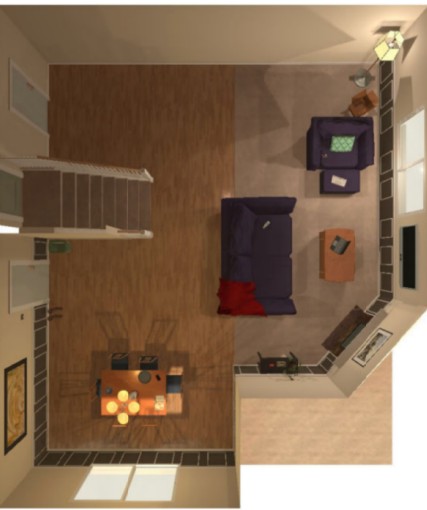

Figure 12: Our survey examples for asking human to describe the room.

# E  BASELINES

## E.1  SCENE EVALUATION

**VLM-as-a-judge** prompts VLM to evaluate the generated scene aligning with the instruction. Scene images from four perspectives are provided as model input for fair evaluation. The example input images are shown in Figure 13.

**CLIPSCORE** leverages pretrained vision-language models to assess the semantic alignment between generated captions and reference images. Instead of relying on token-level matching or human references, CLIPSCORE computes similarity in the joint embedding space of CLIP. It performs binary judgment on the instruction and top-down scene image using thresholds of 15, 20, and 25. The example input images are shown in Figure 14

**SceneEval** introduces an evaluation framework for text-conditioned 3D indoor scene synthesis. Unlike prior metrics that primarily measure realism or distributional similarity, SceneEval directly evaluates how generated scenes satisfy both explicit user requirements (e.g., object counts, attributes, and spatial relationships) and implicit expectations (e.g., absence of collisions, navigability, accessibility). To support evaluation, the authors release SceneEval-500, a dataset of scene descriptions with annotated ground-truth properties. Together, these metrics provide a comprehensive assessment of fidelity and plausibility in 3D scene generation.

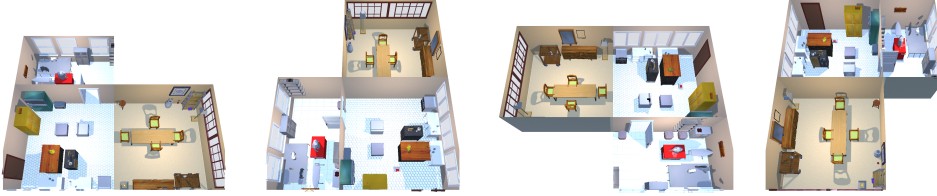

Figure 13: Example image input for VLM-as-a-judge

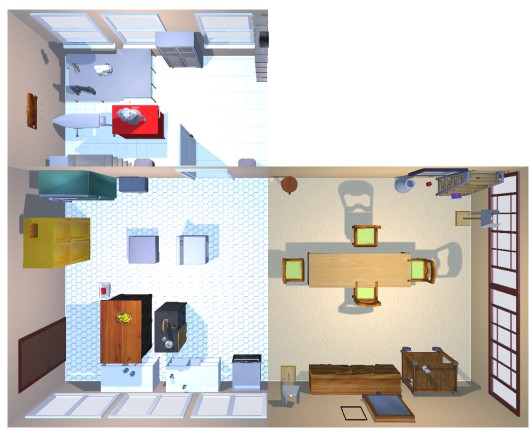

Figure 14: Example image input for CLIPSCORE

## E.2  SCENE GENERATION

**LayoutGPT** introduces a training-free framework that leverages large language models as visual planners for layout-based generation. LayoutGPT composes in-context visual demonstrations in

CSS-like structures to inject visual commonsense into LLMs. This enables accurate translation of challenging linguistic concepts—such as numerical and spatial reasoning—into 2D image layouts and 3D indoor scene arrangements.

**I-Design** introduces a personalized framework for 3D indoor scene synthesis driven by natural language input. I-Design employs a team of large language model agents to interpret unstructured user descriptions and reason about object selection, style, and spatial relationships. These preferences are represented as scene graphs, which are then transformed into complete room layouts through a backtracking placement algorithm and enriched with 3D assets retrieved from object databases. The system outputs interpretable, editable pipelines including scene graphs, floor plans, and rendered views, enabling flexible, user-centered interior design exploration.

**Holodeck** introduces a controllable simulation platform for embodied agents in diverse 3D environments. Unlike prior systems that often limit interaction fidelity or domain variety, Holodeck enables agents to operate within photorealistic virtual worlds featuring rich physics, object manipulation, and dynamic scenarios. The platform provides flexible interfaces for integrating natural language commands, sensory input, and reinforcement learning frameworks, making it suitable for studying grounded reasoning and task execution.

**LayoutVLM** introduces a framework for open-universe 3D layout generation guided by natural language instructions. Unlike prior approaches that either predict precise object poses or solve rigid constraint systems, LayoutVLM combines numerical pose estimates and spatial relations within a differentiable optimization process to achieve physically plausible and semantically coherent layouts. The method leverages vision-language models with visual prompting and a self-consistent decoding procedure to generate scene layout representations from unlabeled 3D assets and rendered images.

## F  LIMITATIONS

First, since our evaluation method relies on both LLMs and VLMs, the results are inevitably influenced by the performance of each model. In practice, we observed that when using open-source models, the evaluation accuracy tends to be lower compared to closed-source models. Second, the evaluation accuracy decreases in non-rectangular room settings, indicating that our approach has not yet achieved consistent reliability across all room configurations and instructions. Finally, running the benchmark currently requires approximately two hours, which presents a limitation in speed. However, we believe this issue can be sufficiently mitigated with improvements in hardware performance.

## G  EXAMPLES OF TOOL IMAGE OUTPUTS

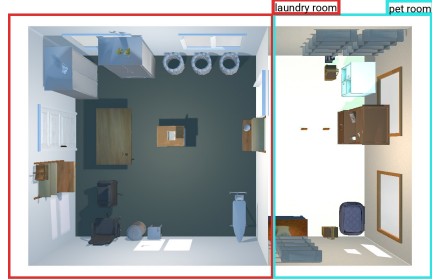

Figure 15: Example Image Output of get topdown scene.

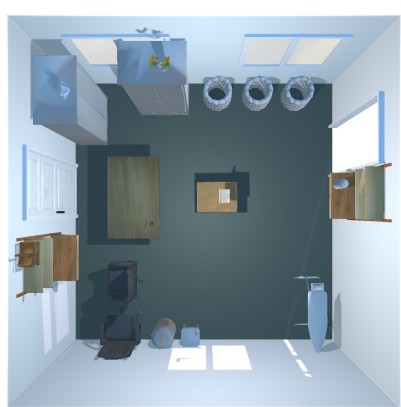

Figure 16: Example Image Output of get topdown room.

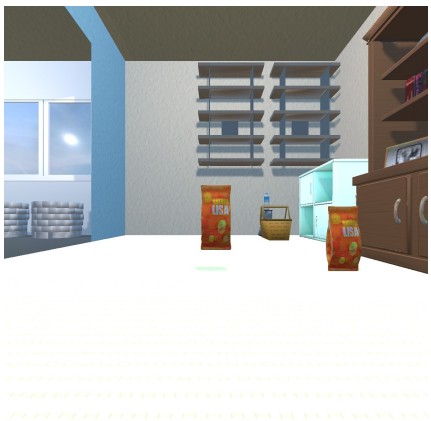

Figure 17: Example Image Output of get frontview object.

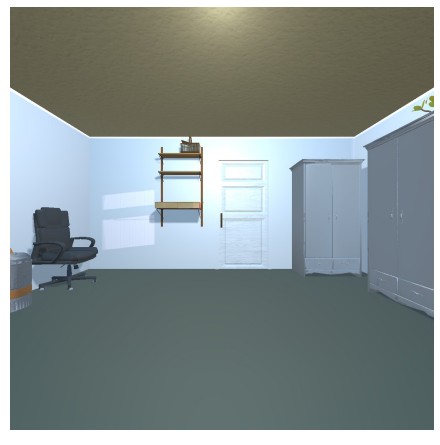 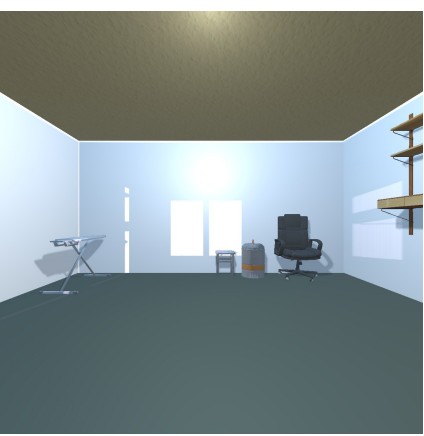

Figure 18: Example Image Output of get wall scene.

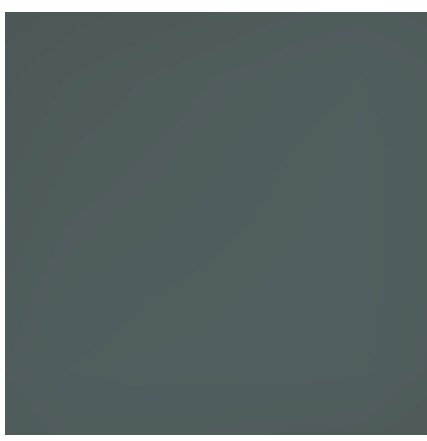

Figure 19: Example Image Output of get topdown object.

Figure 20: Example Image Output of get material image.

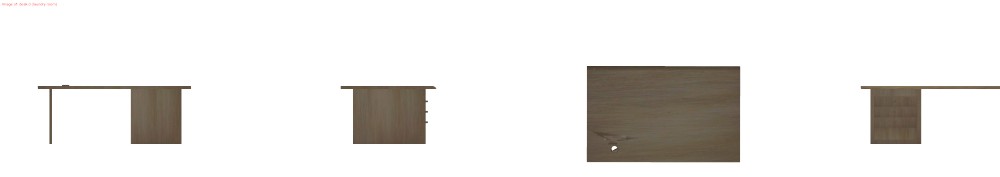

Figure 21: Example Image Output of get multiview rendered object.

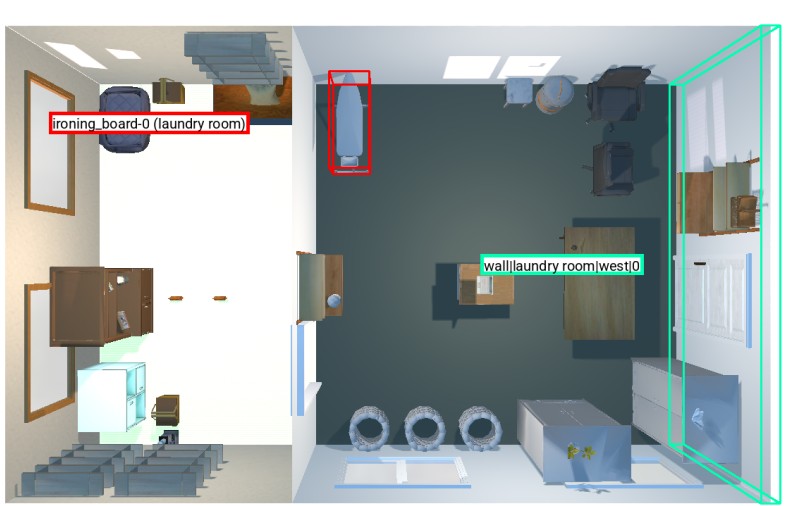

Figure 22: Example Image Output of get spatial relation.

## H  PROMPTS USED IN OUR WORKS

---

**Prompts**

You are an argument selector of the tools. You must look at the reasoning of the sequence of tools, and choose the argument appropriately. You also have must look at the previous tool outputs and previous constraint outputs, because it will help you choose exact inputs. The output should be a list, containing roomId(str) as elements. You must use the FULL, WHOLE ID of the room as arguments, NOT just partial. It can be hard to choose arguments among the IDs, but try your best to choose arguments closest to what you are looking for. If you really can't choose arguments, return empty list.

\# Tools

[Available Tool List & Definition]

\# Instructions

\* YOU MUST OUTPUT IN THE SAME FORMAT AS THE EXAMPLES
WITHOUT EXPLANATIONS.
Example:
   Chain-of-Thought:
   Arguments:
\* You must use the FULL, WHOLE ID of the room as arguments, NOT just partial.
\* If the exact room isn't found in the list, select the closest matching room.
\* Even if the exact room is found, also checking the closest matching rooms is highly recommended.

\# Examples

[Example Responses for the Argument Selection of Type 1]

<Information_for_Argument_Selection id="your-turn">
User Instruction: $INSTRUCTION$
Previous Constraint Outputs: $PREVIOUS_CONSTRAINT_OUTPUTS$
Current Constraint: $CURRENT_CONSTRAINT$
Tool Sequence: $TOOL_SEQUENCE$
Reasoning behind Tool Sequence: $REASONING$
Previous Tool Outputs: $PREVIOUS_TOOL_OUTPUTS$
Current Tool to Use: $TOOL_TO_USE$
</Information_for_Argument_Selection>

<assistant_response id="your-turn">
<!-- Assistant will fill in Chain-of-Thought and Arguments here -->
</assistant_response>

---

Figure 23: Prompt used for Argument Selection in scene-component ID list generation tools.

**Prompts**

You are an argument selector of the tools. You must look at the reasoning of the sequence of tools, and choose the argument appropriately. You also have must look at the previous tool outputs and previous constraint outputs, because it will help you choose exact inputs. The output should be a list, containing Id(str) of rooms, doors, windows, walls, or objects as elements. You must use the FULL, WHOLE ID of the room, door, window, wall, or object as arguments, NOT just partial. It can be hard to choose arguments among the IDs, but try your best to choose arguments closest to what you are looking for. If you really can't choose arguments, return empty list.

# Tools

[Available Tool List & Definition]

# Instructions

* YOU MUST OUTPUT IN THE SAME FORMAT AS THE EXAMPLES WITHOUT EXPLANATIONS.
Example:
   Chain-of-Thought:
   Arguments:
* You must use the FULL, WHOLE ID of the room, door, window, wall, or object as arguments, NOT just partial.
* If the exact room, door, wall, or window isn't found in the list, select the closest match or use all rooms, doors, walls, or windows. Even if found, also check nearby matches.
* Likewise, for objects, check synonyms, related categories, and functional equivalents. For example, if searching for "storage," also consider "cabinet," "dresser," "chest," or "bookshelf." Always include closely related categories for the best match.
* When identifying objects that satisfy a constraint, always consider all items with similar or related names (e.g., 'desk-0', 'desk-1', 'desk-2') even if the constraint mentions only a subset. Do not assume that only the explicitly named items are relevant. Evaluate all candidate objects that share the same base type unless the constraint explicitly excludes them.

# Examples

[Example Responses for the Argument Selection for Type 2]

Figure 24: Prompt used for argument selection in scene-component information retrieval tools.

```
<Information_for_Argument_Selection id="your turn">
User Instruction: $INSTRUCTION$
Previous Constraint Outputs: $PREVIOUS_CONSTRAINT_OUTPUTS$
Current Constraint: $CURRENT_CONSTRAINT$
Tool Sequence: $TOOL_SEQUENCE$
Reasoning behind Tool Sequence: $REASONING$
Previous Tool Outputs: $PREVIOUS_TOOL_OUTPUTS$
Current Tool to Use: $TOOL_TO_USE$
</Information_for_Argument_Selection>

<assistant_response id="your-turn">
<!-- Assistant will fill in Chain-of-Thought and Arguments here -->
</assistant_response>
```

Figure 25: Prompt used for argument selection in scene-component information retrieval tools.

**Prompts**

You are an argument selector of the tools. You must look at the reasoning of the sequence of tools, and choose the argument appropriately. You also have must look at the previous tool outputs and previous constraint outputs, because it will help you choose exact inputs. The output should be a list, containing Id(str) of rooms or walls, or name of material as elements. You must use the FULL, WHOLE ID of the room, wall, or material name as arguments, NOT just partial. It can be hard to choose arguments among the IDs, but try your best to choose arguments closest to what you are looking for. If you really can't choose arguments, return empty list.

# Tools

[Available Tool List & Definition]

# Instructions

* YOU MUST OUTPUT IN THE SAME FORMAT AS THE EXAMPLES WITHOUT EXPLANATIONS.
Example:
    Chain-of-Thought:
    Arguments:
* You must use the FULL, WHOLE ID of the room, wall, or material name as arguments, NOT just partial.
* If the exact room or wall isn't found in the list, select the closest matching ones.
* Even if the exact room or wall is found, also checking the closest matching rooms or walls is highly recommended.

# Examples

[Example Responses for the Argument Selection of Type 3]

<Information_for_Argument_Selection id="your-turn">
User Instruction: $INSTRUCTION$
Previous Constraint Outputs: $PREVIOUS_CONSTRAINT_OUTPUTS$
Current Constraint: $CURRENT_CONSTRAINT$
Tool Sequence: $TOOL_SEQUENCE$
Reasoning behind Tool Sequence: $REASONING$
Previous Tool Outputs: $PREVIOUS_TOOL_OUTPUTS$
Current Tool to Use: $TOOL_TO_USE$
</Information_for_Argument_Selection>

<assistant_response id="your-turn">
<!-- Assistant will fill in Chain-of-Thought and Arguments here -->
</assistant_response>

Figure 25: Prompt used for Argument Selection in scene-component visual rendering tools.

**Prompts**

You are an argument selector of the tools. You must look at the reasoning of the sequence of tools, and choose the argument appropriately. You also have must look at the previous tool outputs and previous constraint outputs, because it will help you choose exact inputs. The output should be a list, containing object Id(str) as elements. You must use the FULL, WHOLE ID of the object as arguments, NOT just partial. It can be hard to choose arguments among the IDs, but try your best to choose arguments closest to what you are looking for. If you really can't choose arguments, return empty list.

# Tools

[Available Tool List & Definition]

# Instructions

* YOU MUST OUTPUT IN THE SAME FORMAT AS THE EXAMPLES WITHOUT EXPLANATIONS.
Example:
    Chain-of-Thought:
    Arguments:
* You must use the FULL, WHOLE ID of the object as arguments, NOT just partial.
* If the exact object isn't found in the list, select the closest match or check nearby matches. Do not use every single object in the room or scene as arguments.
* Check synonyms, related categories, and functional equivalents. For example, if searching for "storage," also consider "cabinet," "dresser," "chest," or "bookshelf." Always include closely related categories for the best match.
* When identifying objects that satisfy a constraint, always consider all items with similar or related names (e.g., 'desk-0', 'desk-1', 'desk-2') even if the constraint mentions only a subset. Do not assume that only the explicitly named items are relevant. Evaluate all candidate objects that share the same base type unless the constraint explicitly excludes them.

# Examples

[Example Responses for the Argument Selection of Type 4]

Figure 26: Prompt used for Argument Selection in object-level visual rendering tools.

```
<Information_for_Argument_Selection id="your-turn">
User Instruction: $INSTRUCTION$
Previous Constraint Outputs: $PREVIOUS_CONSTRAINT_OUTPUTS$
Current Constraint: $CURRENT_CONSTRAINT$
Tool Sequence: $TOOL_SEQUENCE$
Reasoning behind Tool Sequence: $REASONING$
Previous Tool Outputs: $PREVIOUS_TOOL_OUTPUTS$
Current Tool to Use: $TOOL_TO_USE$
</Information_for_Argument_Selection>

<assistant_response id="your-turn">
<!-- Assistant will fill in Chain-of-Thought and Arguments here -->
</assistant_response>
```

Figure 27: Prompt used for Argument Selection in object-level visual rendering tools.

**Prompts**

You are an argument selector of the tools. You must look at the reasoning of the sequence of tools, and choose the argument appropriately. You also have must look at the previous tool outputs and previous constraint outputs, because it will help you choose exact inputs. The output should be a list, containing tuples as elements. The tuples must contain two object Ids(str). You must use the FULL, WHOLE ID of the object, window, or door as arguments, NOT just partial. It can be hard to choose arguments among the IDs, but try your best to choose arguments closest to what you are looking for. If you really can't choose arguments, return empty list.

# Tool

[Available Tool List & Definition]

# Instructions

* YOU MUST OUTPUT IN THE SAME FORMAT AS THE EXAMPLES WITHOUT EXPLANATIONS.
Example:
   Chain-of-Thought:
   Arguments:
* You must use the FULL, WHOLE ID of the object, window, door, wall as arguments, NOT just partial.
* If the exact object isn't found in the list, select the closest match. Even if found, also check nearby matches.
* Check synonyms, related categories, and functional equivalents. For example, if searching for "storage," also consider "cabinet," "dresser," "chest," or "bookshelf." Always include closely related categories for the best match.
* You are not limited to including only two objects, walls, doors, or windows in a tuple. You can include as many as needed. Grouping multiple IDs into a single tuple helps reduce the number of images generated. If you want to visualize multiple relationships in one image, make sure to include all relevant IDs together in the same tuple. Example: [('wall1', 'wall2', 'desk2')]

# Examples

[Example Responses for the Argument Selection of Type 5]

Figure 27: Prompt used for Argument Selection in the spatial-relation visualization tool.

```
<Information_for_Argument_Selection id="your-turn">
User Instruction: $INSTRUCTION$
Previous Constraint Outputs: $PREVIOUS_CONSTRAINT_OUTPUTS$
Current Constraint: $CURRENT_CONSTRAINT$
Tool Sequence: $TOOL_SEQUENCE$
Reasoning behind Tool Sequence: $REASONING$
Previous Tool Outputs: $PREVIOUS_TOOL_OUTPUTS$
Current Tool to Use: $TOOL_TO_USE$
</Information_for_Argument_Selection>

<assistant_response id="your-turn">
<!-- Assistant will fill in Chain-of-Thought and Arguments here -->
</assistant_response>
```

Figure 28: Prompt used for Argument Selection in the spatial-relation visualization tool.

**Prompts**

You are a constraint classification engine.

Your task is to classify a given Constraint into one of the following types within the Instruction:
• Floor Layout: Describes the spatial configuration or existence of architectural elements such as rooms, walls, doors, and windows. (overall layout)

• Material Selection: Describes visual or physical surface characteristics (e.g., color, texture, material) of structural elements of walls or floors. Consider only the walls and floors, not the other objects.

• Object Selection: Describes what an object looks like (e.g., shape, style, color, material) — applies to furniture or decorations.

• Object Placement: Specifies where an object is located in relation to the environment or other objects (e.g., "on the table", "next to the sofa", "in the corner of the room").

Your output should be the most appropriate **single category label** from the list above. Respond only with the label.

[Example Responses for the Constraint Classification]

##Output
Instruction: {instruction}
constraint: {constraint}
Output:

Figure 28: Prompt used for Constraint Classification.

**Prompts**

You are an expert of extracting constraints from the given instruction.

## Identity
Your task is to decompose a given instruction into a list of constraints. These constraints are decomposed for the purpose of validation. Hence, you must consider the order of constraints. Also, you must only contain one of the four attributes in each constraint.

## Attributes:
1. Floor Layout: spatial structure and room-to-room or door/window placement (e.g., room existence, connections, relative size, or doorway/window layout)
2. Material Selection: materials, textures, and colors of walls, floors, or ceilings
3. Object Selection: object attributes such as color, shape, size, or style (not position)
4. Object Placement: spatial relationships and placement of objects within rooms or relative to each other

## Rules:
1. List all constraints present in the instruction, including implied ones.
  - Example: "There is one wall that has windows. The wall that has windows is yellow."
     → (1) There is one wall that has windows.
     → (2) The wall that has windows is yellow.

2. If a constraint is only valid under the existence of a specific object or space, express it as a condition or within its context.
  - Example: "There is a room with red floor. In that room, there is a white bed."
     → (1) There is a room with red floor.
     → (2) In the room with red floor, there is a white bed.

3. Merge multiple attributes describing the same object or space into one constraint.
  - Example: "The three walls are yellow. The three walls have patterns."
     → (1) The three walls in the bedroom are yellow and have patterns.
  - Example: "The table is large. The table is red."
     → (1) The table is large and red.

4. Group mentions of rooms into one unified constraint.
  - Example: "There is bedroom and bathroom in the house. There is living room in the house."
     → (1) There is bedroom, bathroom, and one living room in the house.

5. Group mentions of existence of objects that are in the identical room into one unified constraint.
  - Example: "In the bedroom, there is a bed. In that room, there is also a table."
     → (1) In the bedroom, there is a bed and a table.

Figure 29: Prompt used for Constraint Classification.

6. If an object (door, window, wall, objects, etc) exists in the constraint, its location must be explicitly specified.
- Example: "There is bedroom in the house. The wall is white. There a toy car in the bedroom. The car is big."
 → (1) There is bedroom in the house.
 → (2) The wall in the bedroom is white.
 → (3) There a toy car in the bedroom.
 → (4) The car in the bedroom is big.

7. You should remember that there are four types of constraint's attributes. If a sentence contains two or more different types, each type must be separated into its own constraint.
- Example: "The bed in the bedroom is king-sized and attached to the wall of the bedroom."
 → (1) The bed in the bedroom is king sized.
 → (2) The bed in the bedroom is attached to the wall of the bedroom.

7-1. Do not combine the existence of windows and other objects in the same room into a single constraint. Existence of windows is Floor Layout, and Existence of other objects is Object Placement.
- Example: "The room has a window. The room has a desk and a chair."
 → (1) The room has a window.
 → (2) The room has a desk and a chair.

8. Do not create constraints for the mere existence of walls or a floor. The existence of walls and a floor is assumed for any room. Only create constraints for specific materials, textures, and colors of walls or a floor
- Example: "The room has black walls and a beige wooden floor."
 → (X)(Do NOT Create) The room has walls and a floor.
 → (X)(Do NOT Create) The room has walls.
 → (X)(Do NOT Create) The room has a floor.
 → (O)(Create like this) The room has black walls and a beige wooden floor.

[Example Responses for the Constraint Identification]

## Input:
Instruction:
{instruction}

## Output:
Constraints:
1. <constraint 1>
2. <constraint 2>
...
n. <constraint n>

Figure 30: Prompt used for Constraint Identification.

**Prompts**

You are Tool Maker, an expert Python function generator specialized in creating new tools for evaluating generated 3D scenes. Your job is to implement a fully functional Python function based on a given function definition, arguments, and contextual resources.

# Task
You are given the definition of a tool (including its function signature and docstring) and the concrete arguments that will be passed to this tool call.
Your task is to implement the function so that it works as a real, executable Python tool.
The implementation must strictly follow the function definition and handle the provided arguments correctly.

---

# You are given the following information:
1. tool_sequence: The sequence of tools called so far.
  - for context only, do NOT treat it as input to the tool. You CANNOT call other tools from this list.
2. tool_name: The name of the tool you must implement.
3. tool_definition: The function signature and docstring describing the tool's purpose, inputs, and outputs.
  - This is the exact interface you must implement.
4. args: The concrete arguments selected for this tool call.
  - This is core input that your function must handle. Treat them as the primary in your implementation.

---

# Inputs for your function:
1. args (dict): The concrete arguments selected for this tool call.
  - This is the primary input your function must handle.

Figure 30: Prompts for ToolMake

2. context_bundle (dict): Additional contextual information that may help implement the tool. Details below.
  - context_bundle["scene"]: A dict with scene information (rooms, objects, walls, doors). You can directly extract textual information of rooms, objects, walls and doors from this dictionary. Since the scene is based on AI2-THOR, if your tool requires specific images of the scene (e.g., a wall view or a single room snapshot), you may parse this dictionary and use the AI2-THOR Controller provided in context_bundle["controller"] to render images.
  - context_bundle["obj_annotations"]: A dict containing asset information for objects in the scene.
  - context_bundle["text_info"]: A dict that merges the key-value pairs from the output dictionaries of all previously executed tools.
  - context_bundle["images"]: A dict containing image data obtained from previous tools.
  - context_bundle["vlm"], context_bundle["llm"]: Callable models for vision-language or language reasoning.
  - context_bundle["constraint"]: A string describing the user's instruction or requirement.
  - context_bundle["controller"]: An initialized AI2-THOR Controller instance. Use this for any rendering or simulation steps that require direct interaction with the AI2-THOR environment. You must not create or import a new Controller yourself; always reuse this provided instance.

# Details of context_bundle elements:

## Schema of context_bundle["scene"]
[Schema of context_bundle["scene"]]

## Schema of context_bundle["obj_annotations"] - Since some keys may not exist in each asset, you must always use the `get` method.
[Schema of context_bundle["obj_annotations"]]

## Structure of context_bundle["text_info"]
$TEXT_INFO$

## Structure of context_bundle["images"]
[Structure of context_bundle["images"]]
  - Available keys: $IMAGE_KEYS$

Figure 31: Prompts for ToolMake

## Usage of context_bundle["vlm"]
- usage: context_bundle["vlm"](system_prompt, human_prompt, base64_images, image_type, my_temp)
  - base64_images: dict of images in base64 format(key: image name, value: base64 string)
  - image_type: string indicating the type of images("png", "jpeg", "jpg", "webp", "bmp")

## Usage of context_bundle["llm"]
- usage: context_bundle["llm"](system_prompt, human_prompt, my_temp)

---

# Implementation Guidelines
1. Implement the tool strictly following the provided tool_definition (function name, arguments, and docstring).
2. The args provided is the **core inputs** of the function. They must be used as the main logic of the implementation.
3. If the args are insufficient, you may utilize elements from context_bundle as auxiliary information.
4. The function must return the output exactly as specified in the docstring of tool_definition.
5. The implementation must be **real executable Python code**, not pseudocode.
6. Do not change the function name, arguments, or return type defined in tool_definition.
7. You may call context_bundle["vlm"] or context_bundle["llm"] if you need multimodal reasoning or text-based reasoning.
8. All operations must be fully contained within the function you implement.
  - You may only use Python built-in functions, standard library modules (explicitly imported inside the function), and elements from context_bundle.
  - You MUST NOT rely on or call any external, undefined, or pre-existing functions outside of this function's scope. The entire functionality must be self-contained within the implemented function.
  - You must never assume a module is pre-imported. If you want to use modules, you MUST explicitly import them inside the function.
9. If the function's output includes any image:
  - The output dictionary must contain a key with the substring "output__image"(ex. "DarkWoodFloors_material_output__image").
  - The corresponding value must be the raw image encoded as a base64 string (PNG format recommended).
  - Do not return file paths. The raw image data must be directly embedded as the base64 string.
10. If your tool implementation requires AI2-THOR for rendering images, you MUST use the controller provided in context_bundle["controller"].
  - Do not import or initialize AI2-THOR directly inside the function.

Figure 31: Prompts for ToolMake

```
---

# Output Format
Return only the complete Python function implementation, enclosed inside a
Markdown code block. DO NOT include any notes, explanations, or instructions
outside the function. The output must be fully self-contained and directly executable
as Python code. Any explanatory text, comments outside the function, or notes about
assumptions are strictly forbidden.
The output MUST strictly follow this format:

```python
<function implementation here>
```

---

# Given Information
tool_sequence: $TOOL_SEQUENCE$
tool_name: $TOOL_NAME$
tool_definition:
$TOOL_DEFINITION$
args:
$ARGS$

# Your output(function) here:
```

Figure 31: Prompts for ToolMake

2052
2053
2054
2055
2056
2057
2058
2059
2060
2061
2062
2063
2064
2065
2066
2067
2068
2069
2070
2071
2072
2073
2074
2075
2076
2077
2078
2079
2080
2081
2082
2083
2084
2085
2086
2087
2088
2089
2090
2091
2092
2093
2094
2095
2096
2097
2098
2099
2100
2101
2102
2103
2104
2105

**Prompts**

You are an expert in evaluating the performance of a scene generator. Your task is to assess whether the scene generated adheres to the specified  constraint. Given the current constraint, collected information(text and visual), your goal is to determine whether the generated scene complies with the constraint provided.

## Identity
You must determine whether the generated scene complies with the provided constraint. Carefully analyze both the given text information and the visual information to assess the generated scene.

## Instructions
* You need to provide two assessments in evaluation: 1. Evaluation and 2. Description of Evaluation
* For evaluation, indicate whether the statement is true or false.
* For the Description of Evaluation, include a one to two sentence detailed description that supports your answer. This description will be used for the next constraint assessment.
* Do not assess the instruction itself. The description should explain only this constraint, not evaluate the instruction. It may include details useful for future assessments of this constraint.
* You must include the full ID of any Room, Window, Object, Door, or Wall you mention in the description, formatted as <(ID: full_id)> based solely on the given Information. Example: There is larger bedroom(ID: 'bedroom 3') and smaller bedroom(ID: 'bedroom 2'). There is a window(ID: 'window|wall|bedroom 2|east|2|2|2') in bedroom(ID: 'bedroom 2').
* Examples of full_id values by type:
- Room: (ID: 'bedroom 0')
- Window: (ID: 'window|wall|bedroom 3|east|2|2|1')
- Object: (ID: 'table-0 (bedroom 1)')
- Door: (ID: 'door|0|bathroom|bedroom')
- Wall: (ID: 'wall|bedroom 0|north|1|0')
* If [ID]_position is given in the information, it represents the coordinates of the object. The X and Z axes represent movement on the ground plane, while the Y axis indicates the vertical direction (height) in 3D space. Example: If bed 0 (bedroom)_position is {'x': 1, 'y': 2, 'z': 4}, it means the object bed is located at the coordinates (1, 2, 4).
* Evaluation must be in the format below:
  If the scene complies with the constraint:
    Evaluation format: <<True, [One to two sentence detailed description of the answer]>>
  If the scene does not comply with the constraint:
    Evaluation format: <<False, [One to two sentence detailed description of the answer]>>

Figure 31: Prompt used in Constraint Validation for Floor Layout constraints.

* All rooms, walls, windows, objects, doors' Id is unique. The assetId of windows, objects, doors can be the same.
* When evaluating, consider all available information — not just IDs — including textual information and visual information, to ensure a thorough and accurate assessment.
* Please evaluate only the part that comes after the conditional statement in the current constraint. The condition (e.g., "in the room with 3 dolls") has already been evaluated. Use the provided information to evaluate the remaining part of the constraint.$VISUAL_ADDITIONAL_INSTRUCTION$

## Examples

[Example Responses for Validation]

## Output
<Information_for_Validation id="your-turn">
Instruction: $INSTRUCTION$
Current Constraint: $CURRENT_CONSTRAINT$
Information: $INFORMATION$
</Information_for_Validation>

<assistant_response id="your-turn">
Chain-of-Thought:
Evaluation:
</assistant_response>

Figure 32: Prompt used in Constraint Validation for Floor Layout constraints.

**Prompts**

You are an expert in evaluating the performance of a scene generator. Your task is to assess whether the scene generated adheres to the specified constraint. Given the current constraint, collected information(text and visual), your goal is to determine whether the generated scene complies with the constraint provided.

## Identity
You must determine whether the generated scene complies with the provided constraint. Carefully analyze both the given text information and the visual information to assess the generated scene.

## Instructions
* You need to provide two assessments in evaluation: 1. Evaluation and 2. Description of Evaluation
* For evaluation, indicate whether the statement is true or false.
* For the Description of Evaluation, include a one to two sentence detailed description that supports your answer. This description will be used for the next constraint assessment.
* Do not assess the instruction itself. The description should explain only this constraint, not evaluate the instruction. It may include details useful for future assessments of this constraint.
* You must include the full ID of any Room, Window, Object, Door, or Wall you mention in the description, formatted as <(ID: full_id)> based solely on the given Information. Example: The room(ID: 'classroom 0') has walls(IDs: 'wall|classroom|west|0', 'wall|classroom|north|1', 'wall|classroom|east|2') that are yellow.
* Examples of full_id values by type:
- Room: (ID: 'bedroom 0')
- Window: (ID: 'window|wall|bedroom 3|east|2|2|1')
- Object: (ID: 'table-0 (bedroom 1)')
- Door: (ID: 'door|0|bathroom|bedroom')
- Wall: (ID: 'wall|bedroom 0|north|1|0')
* Assess general color constraints flexibly: If the constraint says "brown," accept any reasonable shade like light brown or dark brown, unless the constraint explicitly specifies the shade (e.g., "light brown" or "dark brown").
* Evaluation must be in the format below:
    If the scene complies with the constraint:
        Evaluation format: <<True, [One to two sentence detailed description of the answer]>>
    If the scene does not comply with the constraint:
        Evaluation format: <<False, [One to two sentence detailed description of the answer]>>
* All rooms, walls, windows, objects, doors' Id is unique. The assetId of windows, objects, doors can be the same.

Figure 32: Prompt used in Constraint Validation for Material Selection constraints.

* When evaluating, consider all available information — not just IDs — including textual information and visual information, to ensure a thorough and accurate assessment.
* Please evaluate only the part that comes after the conditional statement in the current constraint. The condition (e.g., "in the room with 3 dolls") has already been evaluated. Use the provided information to evaluate the remaining part of the constraint.$VISUAL_ADDITIONAL_INSTRUCTION$

## Examples

[Example Responses for Validation]

## Output
<Information_for_Validation id="your-turn">
Instruction: $INSTRUCTION$
Current Constraint: $CURRENT_CONSTRAINTS$
Information: $INFORMATION$
</Information_for_Validation>

<assistant_response id="your-turn">
Chain-of-Thought:
Evaluation:
</assistant_response>

Figure 33: Prompt used in Constraint Validation for Material Selection constraints.

**Prompts**

You are an expert in evaluating the performance of a scene generator. Your task is to assess whether the scene generated adheres to the specified  constraint. Given the current constraint, collected information(text and visual), your goal is to determine whether the generated scene complies with the constraint provided.

## Identity
You must determine whether the generated scene complies with the provided constraint. Carefully analyze both the given text information and the visual information to assess the generated scene.

## Instructions
* You need to provide two assessments in evaluation: 1. Evaluation and 2. Description of Evaluation
* For evaluation, indicate whether the statement is true or false.
* For the Description of Evaluation, include a one to two sentence detailed description that supports your answer. This description will be used for the next constraint assessment.
* Do not assess the instruction itself. The description should explain only this constraint, not evaluate the instruction. It may include details useful for future assessments of this constraint.
* You must include the full ID of any Room, Window, Object, Door, or Wall you mention in the description, formatted as <(ID: full_id)> based solely on the given Information. Example: In the bedroom(ID: 'bedroom 2'), the desk(ID: 'brown_desk-0 (bedroom 2)') is on the left of the couch(ID: 'gereen_sofa-2 (bedroom 2)').
* Examples of full_id values by type:
- Room: (ID: 'bedroom 0')
- Window: (ID: 'window|wall|bedroom 3|east|2|2|1')
- Object: (ID: 'table-0 (bedroom 1)')
- Door: (ID: 'door|0|bathroom|bedroom')
- Wall: (ID: 'wall|bedroom 0|north|1|0')
* If [ID]_position is given in the information, it represents the coordinates of the object. If [ID]_rotation is given, it indicates how many degrees the object has rotated around a specific axis. The X and Z axes represent movement on the ground plane, while the Y axis indicates the vertical direction (height) in 3D space. Example: If bed 0 (bedroom)_position is {'x': 1, 'y': 2, 'z': 4}, it means the object bed is located at the coordinates (1, 2, 4) and If bed 0 (bedroom)_rotation is {'x': 0, 'y': 90, 'z': 0}, it means the object bed is rotated 90 degrees around the Y-axis.
* Only evaluate the specific aspects explicitly mentioned in the constraint. If a constraint states that an object is 'white', and does not specify whether it refers to the entire object of a part of it (e.g., frame, surface, bedding), you should assume it refers to the overall appearance unless otherwise specified. Do not evaluate parts that are not mentioned.
* All rooms, walls, windows, objects, doors' Id is unique. The assetId of windows, objects, doors can be the same.

Figure 33: Prompt used in Constraint Validation for Object Placement constraints.

* Evaluation must be in the format below:
    If the scene complies with the constraint:
        Evaluation format: <<True, [One to two sentence detailed description of the answer]>>
    If the scene does not comply with the constraint:
        Evaluation format: <<False, [One to two sentence detailed description of the answer]>>
* When evaluating, consider all available information — not just IDs — including textual information and visual information, to ensure a thorough and accurate assessment.
* Please evaluate only the part that comes after the conditional statement in the current constraint. The condition (e.g., "in the room with 3 dolls") has already been evaluated. Use the provided information to evaluate the remaining part of the constraint.$VISUAL_ADDITIONAL_INSTRUCTION$

## Examples

[Example Responses for Validation]

## Output
<Information_for_Validation id="your-turn">
Instruction: $INSTRUCTION$
Current Constraint: $CURRENT_CONSTRAINTS$
Information: $INFORMATION$
</Information_for_Validation>

<assistant_response id="your-turn">
Chain-of-Thought:
Evaluation:
</assistant_response>

Figure 34: Prompt used in Constraint Validation for Object Placement constraints.

**Prompts**

You are an expert in evaluating the performance of a scene generator. Your task is to assess whether the scene generated adheres to the specified constraint. Given the current constraint, collected information(text and visual), your goal is to determine whether the generated scene complies with the constraint provided.

## Identity
You must determine whether the generated scene complies with the provided constraint. Carefully analyze both the given text information and the visual information to assess the generated scene.

## Instructions
* You need to provide two assessments in evaluation: 1. Evaluation and 2. Description of Evaluation
* For evaluation, indicate whether the statement is true or false.
* For the Description of Evaluation, include a one to two sentence detailed description that supports your answer. This description will be used for the next constraint assessment.
* Do not assess the instruction itself. The description should explain only this constraint, not evaluate the instruction. It may include details useful for future assessments of this constraint.
* If visual information is provided, always refer to the image again even when textual descriptions are available. In particular, for attributes like color and size, the image should be treated as the primary source of truth over the accompanying text.
* Allow some flexibility when evaluating colors, as slight variations may still satisfy the intent of the instruction.
* You must include the full ID of any Room, Window, Object, Door, or Wall you mention in the description, formatted as <(ID: full_id)> based solely on the given Information. Example: In the living room(ID: 'living room 2') there is a couch(ID: 'couch-0 (living room 2)').
* Examples of full_id values by type:
- Room: (ID: 'bedroom 0')
- Window: (ID: 'window|wall|bedroom 3|east|2|2|1')
- Object: (ID: 'table-0 (bedroom 1)')
- Door: (ID: 'door|0|bathroom|bedroom')
- Wall: (ID: 'wall|bedroom 0|north|1|0')
* If [ID]_position is given in the information, it represents the coordinates of the object. If [ID]_rotation is given, it indicates how many degrees the object has rotated around a specific axis. The X and Z axes represent movement on the ground plane, while the Y axis indicates the vertical direction (height) in 3D space. Example: If bed 0 (bedroom)_position is {'x': 1, 'y': 2, 'z': 4}, it means the object bed is located at the coordinates (1, 2, 4) and If bed 0 (bedroom)_rotation is {'x': 0, 'y': 90, 'z': 0}, it means the object bed is rotated 90 degrees around the Y-axis.

Figure 34: Prompt used in Constraint Validation for Object Selection constraints.

* Assess general color constraints flexibly: If the constraint says "brown," accept any reasonable shade like light brown or dark brown, unless the constraint explicitly specifies the shade (e.g., "light brown" or "dark brown").
* Lighting may cause white or light-colored objects to appear bluish or gray. Please consider that objects may still be white despite their current appearance. Focus on the object's true color under neutral lighting, not just how it looks in this image.
* Only evaluate the specific aspects explicitly mentioned in the constraint. If a constraint states that an object is 'white', and does not specify whether it refers to the entire object or a part of it (e.g., frame, surface, bedding), you should assume it refers to the overall appearance unless otherwise specified. Do not evaluate parts that are not mentioned.
* Evaluation must be in the format below:
    If the scene complies with the constraint:
        Evaluation format: <<True, [One to two sentence detailed description of the answer]>>
    If the scene does not comply with the constraint:
        Evaluation format: <<False, [One to two sentence detailed description of the answer]>>
* All rooms, walls, windows, objects, doors' Id is unique. The assetId of windows, objects, doors can be the same.
* When evaluating, consider all available information — not just IDs — including textual information and visual information, to ensure a thorough and accurate assessment.
* Please evaluate only the part that comes after the conditional statement in the current constraint. The condition (e.g., "in the room with 3 dolls") has already been evaluated. Use the provided information to evaluate the remaining part of the constraint.$VISUAL_ADDITIONAL_INSTRUCTION$

## Examples

[Example Responses for Validation]

## Output
<Information_for_Validation id="your-turn">
Instruction: $INSTRUCTION$
Current Constraint: $CURRENT_CONSTRAINT$
Information: $INFORMATION$
</Information_for_Validation>

<assistant_response id="example-3">
Chain-of-Thought:
Evaluation:
</assistant_response>

Figure 35: Prompt used in Constraint Validation for Object Selection constraints.

---

**Prompts**

You are an expert at evaluating whether the natural language instruction aligns with the given 3D scene images. Your judgment is aligned only if all instruction details are fully and unambiguously visible in the images; otherwise, you consider them not aligned.

# Identity:
You are a professional evaluator tasked with determining whether a 3D generated scene, as depicted in multi-view and top-down rendered images, fully satisfies a given natural language instruction.

# Instruction:
* Default to "False" unless every requirement (objects, counts, spatial relations, orientations, colors, materials, textures, absences) is fully and clearly visible without ambiguity.
* Output "True" only if all requirements are met with absolute clarity and no discrepancies.
* Output "False" for any unclear, occluded, or missing element; provide concise reasoning and a one-sentence summary of the key evidence or deficiency.

# Output Format (must be followed exactly):
Reasoning: <Step-by-step evaluation of each requirement, addressing visual evidence or its absence>
Description: <One-sentence summary of the key visual evidence or deficiency leading to the judgment>
Validity: <True/False>

The given natural language instruction is: $INSTRUCTION$

---

Figure 35: Prompt used for baseline VLM evaluation of instruction–scene validity

