# OpenReview forum: "LEGO-Eval: Towards Fine-grained Evaluation on Synthesizing 3D Embodied Environments with Tool Augmentation"
_ICLR.cc/2026/Conference — Submitted to ICLR 2026_

### Official Review · Reviewer_oMe6 · 2025-11-01

**Soundness:** 3
**Presentation:** 3
**Contribution:** 2
**Rating:** 6
**Confidence:** 3

**Summary:**

This paper proposes a new metric, LEGO-Eval, for evaluating the 3D embodied environments.  LEGO-Eval uses multiple tools to extract the information from the scene and verify if they satisfy the constraints. The evaluation results demonstrate LEGO-Eval achieves much higher agreement with human judger than previous metrics like CLIP-Score.

**Strengths:**

1. The proposed metric has much better agreement with human judgement compared to other metrics.
2. The refinement experiments is a highlight that such metrics can be reliable rewards for improving systems.

**Weaknesses:**

**Require dense scene annotations.** This is the major concern of the proposed evaluation metric which requires dense annotations of each assets (attributes, locations, etc). However, for some methods which generate entire scene in a single mesh (e.g., diffusion-based model), the proposed method cannot use tools to get those information. The results in Table 2 also validate this concern that, with textual reasoning on attributes and precise spatial information from rendering engine, the performance will drop significantly. Therefore, the usability of the proposed metric is quite limited.

**Questions:**

1. How many human annotators are recruited for collecting human judgments?
2. How many examples are used in Figure 7 to get the results?

---

> ### Author Response · Authors · 2025-11-26
> **Response to Reviewer oMe6**
>
> Dear Reviewer oMe6,
> We appreciate your comments and positive feedback on our work. We will address the concerns and the questions raised by the reviewer in the comments below.
>
> ### **W. Dense scene annotations**
> Thank you for raising the concern about LEGO-Eval’s dependency on dense scene annotations. We have provided clarifications regarding this issue in the general response.
>
> ### **Q1. Human judgments**
> 5 human annotators were recruited for collecting human judgments. Each annotator annotated approximately 30 scenes.
>
> ### **Q2. Refinement experiment**
> 130 instructions, LEGO-Bench, were used to get the results in Figure 7\.

---

### Official Review · Reviewer_MKU2 · 2025-11-01

**Soundness:** 3
**Presentation:** 3
**Contribution:** 4
**Rating:** 6
**Confidence:** 4

**Summary:**

The paper tackles a pain point in text-guided 3D scene generation: we can now generate scenes from language, but we can't reliably tell if the scene actually matches the detailed instruction. Existing automatic evaluators (e.g., CLIPScore) don't really understand 3D layouts, and they crumble on constraints.

To address this, the authors propose LEGO-EVAL, a tool-augmented evaluation pipeline. The idea is: Take the long instruction, identify and break it into structured constraints (4 types so far); For each constraint, plan which tools to call (Unity environment interaction, textual reasoning, VLM reasoning), execute those tools to actually ground the entities, and then give a binary judgment with evaluation explanations, declare the whole scene valid only if all constraints pass.

Alongside this, the authors build LEGO-BENCH, focusing on the attributes and spatial relationships of 3D scene generations, so that different evaluators can be compared.

**Strengths:**

- Reframes 3D-scene evaluation as a tool-augmented reasoning task. Combining constraint extraction, planning, and multimodal tool calls for grounding is a novel and well-motivated contribution.
- The pipeline and tool taxonomy are well-explained. Figures and examples make the method intuitive.
- Strong experiments with fair baselines (e.g., CLIPScore, SceneEval). Clear metrics, ablations, and human alignment analyses.

**Weaknesses:**

- Simulator dependency: LEGO-EVAL assumes access to the scene graph and Unity backend. This may not be available in many real settings like photorealistic assets.
- Scene limination: LEGO-BENCH is limited to indoor scenes. Broader or more varied data would strengthen claims.
- Failure analysis: It's unclear which constraint types cause most errors for baselines.

**Questions:**

1. Can LEGO-EVAL operate without simulator access?
2. What is the average tool-call cost per instruction and runtime per scene?

---

> ### Author Response · Authors · 2025-11-26
> **Response to Reviewer MKU2**
>
> Dear Reviewer MKU2,
> We appreciate the reviewer’s thoughtful feedback and recognition of our contributions. We address the concerns you have raised in the comments below:
>
> ### **W1 (Q1). Simulator & scene graph dependency**
> Thank you for highlighting this valuable aspect of our approach. We have clarified LEGO-Eval’s dependency on the simulator and scene graphs in the general response.
>
> ### **W2. Scene limitation to indoor settings**
> We appreciate the reviewer’s observation. LEGO-BENCH is indeed built upon the AI2-THOR simulator, which is currently limited to indoor environments. However, we would like to emphasize that our benchmark includes diverse indoor scenes beyond standard room types.
>
> Specifically, our instruction set spans not only kitchens, bedrooms, bathrooms, and living rooms, but also varied environments such as cafeterias, offices, and even a space museum. We believe this diversity meaningfully captures a broad spectrum of real-world indoor settings and provides a strong foundation for evaluating generalization within indoor domains.
>
> ### **W3. Failure analysis**
>
> **Error analysis by constraint type**
> We conducted an error analysis of VLM-as-a-Judge by computing, for each constraint type, the proportion of incorrectly judged cases out of the total number of evaluated constraints.
>
> VLM-as-a-Judge misjudged 50% of floor layout cases (136 errors), 73% of material selection cases (140 errors), 81% of object selection cases (235 errors), and 69% of object placement cases (341 errors). These results indicate that object selection, material selection, and object placement constraints are challenging for VLM-as-a-Judge.
>
> **Error analysis by failure mode**
> We have also conducted error analysis on VLM-as-a-Judge by classifying their type of errors. Specifically, we randomly sampled a total of 100 instructions (i.e., 1,058 constraints), with 50 labeled as true and 50 as false, and analyzed the failure modes across several baseline methods. The failure modes were categorized into:
>
> * **Identification error**: Failure to correctly locate or identify scene elements.
> * **Spatial error**: Failure to accurately infer spatial relationships between scene elements.
> * **Attribute error**: Failure to correctly infer properties such as color, size, or shape of scene elements.
>
> Out of 367 errors, there were 150 errors for both identification error and attribute error. The remaining 67 were spatial error. This reveals that failures often occur in the early stages of visual grounding, particularly in grounding scene components.
>
> LEGO-Eval performs multi-hop grounding by explicitly planning which tools to use and how to use them in order to extract the necessary information for constraint evaluation. This structured approach enables more reliable and interpretable assessments compared to end-to-end judgments by VLMs.
>
> ### **Q2. Cost tradeoff comparison**
> We conducted a cost trade-off analysis. The results are provided in the general response.

---

### Official Review · Reviewer_zAu4 · 2025-11-04

**Soundness:** 3
**Presentation:** 3
**Contribution:** 3
**Rating:** 4
**Confidence:** 3

**Summary:**

This paper builds on the core insight that existing evaluation methods (such as using VLMs as judges) do not adequately match fine-grained text instructions with 3D scenes; this becomes a problem for downstream use-cases such as text-to-scene synthesis. To address this, the paper introduces (i) LEGO-Bench, a manually annotated (n = 130) dataset of text-scene pairs, and (ii) LEGO-Eval, a tool-based evaluation method that drastically outperforms VLM-as-judges when compared to ground-truth.

**Strengths:**

1. LEGO-Eval’s tool-grounded pipeline drives a striking jump in F1 versus the usual VLM-as-judge baselines, showing that explicit grounding leads to better alignment verdicts.

1. LEGO-Bench is valuable: 130 instructions with roughly 1.2k hand-checked constraints covering both architectural makeup and object relations give the community a realistic, fine-grained stress test. The field of scene graphs, while tangential to this paper, _also_ incidentally lacks high-quality fine-grained annotations for scenes, despite it being a common drawback of VLMs.

1. The paper is well-written and easy to follow. The experimental coverage is thoughtful; it has ablations over tool types, comparisons against four synthesis systems, and the Holodeck refinement loop (Fig. 7) all help illustrate the usefulness of LEGO-Eval/Bench.

**Weaknesses:**

1. The paper does not provide conclusive evidence (or even a brief discussion) to the claim that finer-grained text-scene alignment leads to real embodied gains. The paper does provide _preliminary_ evidence via the Holodeck refinement vignette (Fig. 7); however there’s no “detect -> repair -> retrain” loop or even a pointer to existing sim-to-real failures. A minimal downstream study (or stronger citations) would make the story much more convincing.


1. LEGO-Eval leans on several Unity-facing tools, so the comparison to image-only VLM judges risks being apples-to-oranges. Please add baselines that ingest similar structure (e.g., VLM + detector/scene-graph outputs, see weakness #2 and question #3 below) to show the lift truly comes from the proposed orchestration.


3. The paper has a limited analysis of failure modes of VLM-as-judges. Figure 8 hints that VLM judges mostly hallucinate or misidentify objects, yet we never see how often that happens or how severe it is. Please tally the dominant error types (mis-identification, spatial mistakes, attribute mismatches) so we can tell whether cascading failures are the main culprit. If mis-identification dominates, test baselines that feed object detection outputs [1,2] or structured summaries from scene graph generators to see how much ground they recover. Likewise, benchmark 3D-language models (3D-LLM, Point-LLM) that already encode volumetric context. A quantitative breakdown plus these stronger baselines would clarify when LEGO-Eval is indispensable versus when richer perception priors nearly match it.

[1] Grounding-DINO. Grounding dino: Marrying dino with grounded pre-training for open-set object detection. https://arxiv.org/abs/2303.05499
[2] YoloV8, https://yolov8.com/

**Questions:**

1. As mentioned above, do you have any downstream evidence (even a small detect -> repair -> retrain study, or at least documented cases in the literature) that tighter instruction-scene alignment boosts embodied performance?

1. Can you quantify the cost tradeoff of LEGO-Eval: number of tool calls per instruction, duration (limitation currently mentions "two hours" for the 260 samples), and approximate compute cost per sample. How do these figures compare to the single-pass VLM judge?

1. It seems that LEGO-Eval is specifically targeted at static, top-down scenes and requires careful tool curation. Can it theoretically cope with dynamic scenes? Furthermore, would it make sense to add scene graph generators [1,2], which produce fine-grained annotations from scenes, as a possible baseline?

I am quite willing to raise my score if the above questions and weaknesses are addressed.

[1] Gu et al, ConceptGraphs: Open-Vocabulary 3D Scene Graphs for Perception and Planning. https://arxiv.org/abs/2309.16650
[2] Huang et al, LASER: A Neuro-Symbolic Framework for Learning Spatial-Temporal Scene Graphs with Weak Supervision. https://arxiv.org/abs/2304.07647

---

> ### Author Response · Authors · 2025-11-26
> **Response to Reviewer zAu4 (Part 1)**
>
> Dear Reviewer zAu4,
> We sincerely appreciate your understanding of our contribution. Your constructive feedback has been invaluable in helping us identify areas for improvement.
>
> ### **W1 (Q1). Instruction-scene alignment boosts embodied performance**
>
> We understand the reviewer’s concern that we did not provide conclusive evidence that directly demonstrates the impact of instruction-scene alignment on embodied performance. However, we believe that tighter instruction-scene alignment leads to better embodied performance, as supported by prior work.
>
> HSSD-200 \[1\] demonstrates that training on realistic synthetic scenes is more effective for real-world task performance than training on a larger number of less realistic ones:
>
> * Embodied agents were trained on the ObjectNav task using two sources: Procthor, which often produces unrealistic layouts, and HSSD-200, which provides human-annotated, realistic scenes.
> * Procthor \[2\] offered 10,000 scenes, while HSSD-200 used only 120\. Despite this 100× difference in scale, agents trained in HSSD-200 achieved better zero-shot transfer performance in photorealistic environments such as MP3D \[3\] and HM3DSem \[4\].
>
> These findings suggest that instruction-scene alignment \- rather than data volume alone \- plays a critical role in downstream embodied performance and sim-to-real transfer. Based on this, when scenes are generated to closely align with given instructions, it becomes possible to leverage detailed real-world descriptions to automatically construct realistic and semantically grounded synthetic environments. We believe that such environments can facilitate more effective training and enhance sim-to-real transfer for embodied agents.
>
> ### **W2 (Q3). Additional baselines & dynamic environment evaluation**
>
> **Additional baselines**
>
> Following the reviewer’s suggestion, we have added five additional baselines that incorporate scene information, making them more comparable to LEGO-Eval’s use of Unity-derived structured representations.
>
> * **Setup.** First, we employed Grounding DINO \[5\] to annotate all input images with bounding boxes, and used these annotated images as input to the VLMs. Second, we constructed scene graphs using scene graph-based methods \[6,7\] and incorporated the generated graphs into the prompts, alongside the original image inputs, for the VLMs. We used GPT-4.1 as the base model. Third, we converted the scenes from structured text into point cloud representations to support 3D Language models \[8,9\].
> * **Results.** Despite providing additional contextual information to the LLMs, the results show that LEGO-Eval consistently outperforms these enhanced baselines in both the Partial and Holistic evaluation settings. The results can be found in the table below.
>
> | Method | Holistic ↑ |  | Partial ↑ |  |
> | :---: | :---: | :---: | :---: | :---: |
> |  | **F1** | **Cohen’s Kappa** | **F1** | **Cohen’s Kappa** |
> | VLM-as-a-Judge | 0.40 | 0.05 | 0.68 | 0.35 |
> | Grounding Dino \[5\] | 0.37 | 0.04 | 0.58 | 0.24 |
> | ConceptGraphs \[6\] | 0.36 | 0.02 | 0.43 | 0.12 |
> | LASER \[7\] | 0.37 | 0.03 | 0.49 | 0.16 |
> | 3D-LLM \[8\] | 0.44 | 0.00 | 0.45 | 0.00 |
> | Point-LLM \[9\] | 0.38 | 0.04 | 0.50 | 0.03 |
> | **LEGO-Eval** | **0.81** | **0.63** | **0.83** | **0.66** |
>
> **Dynamic environment evaluation**
>
> Thank you for your thoughtful question. While our current evaluation primarily focuses on static scenes, LEGO-Eval is designed to support flexible addition and replacement of tools, which allows for theoretical extensibility to dynamic environments as well.
>
> Specifically, new evaluation tools can be developed and seamlessly integrated to handle different types of scenes. For example, if dynamic object states need to be evaluated, a tracking tool can be created and plugged into the framework to capture temporal changes. Therefore, although not demonstrated in this work, LEGO-Eval is theoretically capable of supporting dynamic scene evaluation through appropriate tool extensions.

---

> ### Author Response · Authors · 2025-11-26
> **Response to Reviewer zAu4 (Part 2)**
>
> ### **W3. Failure modes type count**
>
> We conducted a qualitative and quantitative investigation of the dominant error types. Specifically, we randomly sampled a total of 100 instructions (i.e., 1,058 constraints), with 50 labeled as true and 50 as false, and analyzed the failure modes across several baseline methods. The failure modes were categorized into:
>
> * **Identification error**: Failure to correctly locate or identify scene elements.
> * **Spatial error**: Failure to accurately infer spatial relationships between scene elements.
> * **Attribute error**: Failure to correctly infer properties such as color, size, or shape of scene elements.
>
> Surprisingly, we found that adding more input information did not reduce the error rate. In fact, identification errors often increased:
>
> * **Grounding DINO.** The model frequently produced inaccurate bounding boxes, and the VLM-as-a-Judge placed undue weight on these incorrect detections, leading to misjudgments.
> * **Scene graph inputs.** The generated scene graphs were often inaccurate or omitted critical information, yet the VLM-as-a-Judge treated them as highly reliable, amplifying evaluation errors.
> * **3D-language models.** Due to input size limitations, more than half of the point cloud had to be discarded, resulting in a loss of important spatial and semantic detail and thus an incomplete understanding of the scene.
>
> The result of the error count can be found in the table below:
>
> | Method | Total Error ↓ | Identification Error ↓ | Spatial Error ↓ | Attribute Error ↓ |
> | :---: | :---: | :---: | :---: | :---: |
> | VLM-as-a-Judge | 367 | 150 | 76 | 150 |
> | Grounding Dino \[5\] | 451 | 288 | 42 | 121 |
> | ConceptGraphs \[6\] | 644 | 395 | 79 | 170 |
> | LASER \[7\] | 497 | 233 | 100 | 164 |
> | 3D-LLM \[8\] | 271 | 192 | 24 | 55 |
> | Point-LLM \[9\] | 447 | 203 | 72 | 172 |
> | **LEGO-Eval (Ours)** | **119** | **53** | **20** | **46** |
>
> LEGO-Eval, in contrast, explicitly considers what information is necessary for a reliable evaluation during its tool planning stage. Based on this reasoning, it selects and sequences tools that can provide the required information. This enables LEGO-Eval to gather accurate and contextually appropriate information for each instruction, resulting in evaluations that are significantly more robust and reliable.
>
> ### **Q2. Cost Tradeoff Comparison**
> We appreciate the reviewers for highlighting this important comparison. In response, we conducted the suggested cost trade-off analysis and present the results in this general response.
>
> **References**
>
> \[1\] Khanna, Mukul, et al. "Habitat synthetic scenes dataset (hssd-200): An analysis of 3d scene scale and realism tradeoffs for objectgoal navigation." *Proceedings of the IEEE/CVF Conference on Computer Vision and Pattern Recognition*. 2024., [https://arxiv.org/abs/2306.11290](https://arxiv.org/abs/2306.11290)
> \[2\] Deitke, Matt, et al. "🏘️ ProcTHOR: Large-Scale Embodied AI Using Procedural Generation." *Advances in Neural Information Processing Systems* 35 (2022): 5982-5994., https://arxiv.org/abs/2206.06994
> \[3\] Chang, Angel, et al. "Matterport3D: Learning from RGB-D Data in Indoor Environments." *2017 International Conference on 3D Vision (3DV)*. IEEE Computer Society, 2017., https://arxiv.org/abs/1709.06158
> \[4\] Yadav, Karmesh, et al. "Habitat-matterport 3d semantics dataset." *Proceedings of the IEEE/CVF Conference on Computer Vision and Pattern Recognition*. 2023., https://arxiv.org/abs/2210.05633
> \[5\] Liu, Shilong, et al. "Grounding dino: Marrying dino with grounded pre-training for open-set object detection." *European conference on computer vision*. Cham: Springer Nature Switzerland, 2024., https://arxiv.org/abs/2303.05499
> \[6\] Gu, Qiao, et al. "Conceptgraphs: Open-vocabulary 3d scene graphs for perception and planning." *2024 IEEE International Conference on Robotics and Automation (ICRA)*. IEEE, 2024., https://arxiv.org/abs/2309.16650
> \[7\] Huang, Jiani, et al. "Laser: A neuro-symbolic framework for learning spatial-temporal scene graphs with weak supervision." *arXiv preprint arXiv:2304.07647* (2023)., https://arxiv.org/abs/2304.07647
> \[8\] Hong, Yining, et al. "3d-llm: Injecting the 3d world into large language models." *Advances in Neural Information Processing Systems* 36 (2023): 20482-20494., https://arxiv.org/abs/2307.12981
> \[9\] Guo, Ziyu, et al. "Point-bind & point-llm: Aligning point cloud with multi-modality for 3d understanding, generation, and instruction following." *arXiv preprint arXiv:2309.00615* (2023)., https://arxiv.org/abs/2309.00615

---

### Official Review · Reviewer_5Zma · 2025-11-04

**Soundness:** 3
**Presentation:** 3
**Contribution:** 3
**Rating:** 2
**Confidence:** 4

**Summary:**

The paper introduces LEGO-EVAL, a new evaluation framework that uses a diverse set of tools (for environment interaction, textual reasoning, and multimodal reasoning). This tool-augmented approach allows it to explicitly ground scene components and accurately assess if the generated 3D scene aligns with complex, detailed instructions.

The authors also created LEGO-BENCH, a new benchmark of fine-grained instructions for 3D environments. Experiments show LEGO-EVAL dramatically outperforms VLM-as-a-judge (0.81 vs. 0.40 F1 score) in alignment with human judgments.

**Strengths:**

Instead of relying on one AI model to just "look" at the scene, the paper introduces LEGO-EVAL, which acts more like a detective. It uses a set of specialized "tools" to check specific facts—one tool to find all the objects, another to check their color, and another to measure their spatial relationships.

The authors created their own difficult test (called LEGO-BENCH) full of complex instructions. They proved their new "judge" (LEGO-EVAL) is far more accurate than older methods.

Experiments also show that current AI models for building 3D scenes are still very bad at following detailed instructions, failing most of the time.

**Weaknesses:**

The paper introduces a new test set called LEGO-BENCH, but it only contains 130 instructions. This is a very small number, which might not be enough to prove the necessity of making such a benchmark. In fact, there are many indoor scene synthesis benchmarks and it is not even worthwhile to start a new language-instructure synthesis from scratch.

In LEGO-BENCH, the scenes used to test the evaluator were created "manually." This process is very slow, expensive, and hard to scale. Utilizing a sequence call of LLM APIs to generate, verify, and refine, seemed to be costly and super inefficient in generating a simple contraints scene from natural language.

The best results come from using "GPT-4.1." The paper shows that performance drops significantly when using smaller or different models. This means the system's success isn't just its smart design but also its reliance on a very powerful (and expensive) "brain" that not everyone can access or afford. For example, does a 7B or 4B model good enough to generate good results based on the method proposed?

Considering the downstream tasks, what can this method bring advantages to? e.g. robotic learning? navigation? gaming?  The dataset from "manually collect instructions for 3D scene synthesis" may not be super useful for other envs, tasks, game engines, simulations, e.t.c. In another language, the impact of this LEGO-BENCH is too small.

**Questions:**

See weaknesses

---

> ### Author Response · Authors · 2025-11-26
> **Response to Reviewer 5Zma (Part 1)**
>
> Dear Reviewer 5Zma,
> We appreciate your thoughtful review. We hope that the following responses will help clarify the importance of LEGO-Eval and LEGO-Bench, and their impacts on the research community:
>
> ### **W1. Limitations of LEGO-Bench**
> **Comparable scale of LEGO-Bench with existing benchmarks**
> While LEGO-Bench provides 130 instruction-scene pairs, we note that other text-to-3D benchmarks have made a significant impact in the research community with a comparable amount of test data—such as DreamFusion \[1\] with 153 instructions and Eval3D \[2\] with 160\.
>
> **Limitations of existing benchmarks for 3D scene synthesis**
> Although there are many indoor 3D scene synthesis datasets, existing datasets have limitations for the usage of LLM-based scene synthesis methods:
>
> * Most text-to-3D datasets are for mesh or point cloud of 3D scenes, while LLM-based methods output scenes in textual representations (i.e., JSON).
> * The dataset for LLM-based scene generation methods only provides instructions and not scenes, limiting its applicability.
> * LEGO-Bench provides both detailed instructions and aligned 3D scenes, allowing broader usage such as training or evaluating reward models for text-to-3D scene generation.
>
>
> We summarize the comparison in the table below:
>
> | Text-to-3D Benchmark/Dataset | Textual Scene Representation | Text-aligned Scenes |
> | :---: | :---: | :---: |
> | InstructionScene \[3\] | ❌ (Mesh) | ✅ |
> | SceneVerse \[4\] | ❌ (PointCloud) | ✅ |
> | SceneEval \[5\] | ✅ | ❌ |
> | **LEGO-Bench (Ours)** | ✅ | ✅ |
>
> **Faithful evaluation is essential for benchmark reliability**
> Crucially, the reliability of a benchmark does not depend solely on its size but also on how accurately it supports evaluation. As the number of instructions increases, human evaluation becomes impractical, making an accurate and scalable automated evaluation essential. However, existing benchmarks either lack evaluation methodologies altogether \[3,4\] or rely on metrics with limited accuracy \[5\]. LEGO-Bench addresses this gap through LEGO-Eval—a robust evaluation framework that significantly outperforms existing methods. This strong evaluation capability is central to the trustworthiness of the benchmark, enabling accurate assessment of instruction-scene alignment.
>
>
> ### **W2. Manually annotating scenes is too costly**
> As noted above, prior benchmarks do not provide text-aligned scene datasets that can be used or extended for LLM-based methods. Moreover, as shown in Table 3, existing LLM-based scene synthesis methods still struggle to generate scenes that are fully aligned with a given instruction. Because no suitable dataset exists and automatic generation is not yet reliable, we manually collected instruction-aligned scenes. Although each annotator spent approximately 3-4 days collecting scenes, it allowed us to construct 130 instruction-aligned scenes—a resource that the community previously lacked.
>
>
> ### **W3. Reliance on the base model performance**
> **Evaluation performance reflects the tool planning capability**
> We understand the reviewer’s concern regarding the reliance on stronger models. As shown in Section 5 (Table 5), LEGO-Eval’s evaluation performance is closely tied to the base model’s tool planning capability—its ability to identify the necessary information and select appropriate tools to retrieve it. Thus, we find models with better planning abilities to show higher evaluation capabilities. Consequently, models with stronger tool planning capabilities demonstrate higher evaluation performance.
>
> The tool selection performance of each base model can be found in the table below:
>
> | Models | Tool F1 ↑ | GED ↓ |
> | :---: | :---: | :---: |
> | Qwen2.5VL-32b | 0.57 | 2.87 |
> | GPT4.1-mini | 0.70 | 2.30 |
> | Qwen3VL-32b | 0.71 | 1.79 |
> | GPT4.1 | **0.77** | **1.35** |
>
> **LEGO-Eval remains effective in resource-constrained settings**
> Importantly, we also observe that when using smaller models such as 4B or 8B, LEGO-Eval still better aligns with human judgment than VLM-as-a-Judge. This suggests that the framework itself is effective even without relying on the most powerful models. This makes the method practical in settings where API usage or computational resources are constrained.
>
> The comparison between evaluation methods can be found in the table below:
>
> | Method | Holistic ↑ |  | Partial ↑ |  |
> | :---: | :---: | :---: | :---: | :---: |
> |  | **F1** | **Cohen’s Kappa** | **F1** | **Cohen’s Kappa** |
> | VLM-as-a-Judge (Qwen3VL-4b) | 0.37 | 0.02 | 0.46 | 0.11 |
> | VLM-as-a-Judge (Qwen3VL-8b) | 0.35 | 0.02 | 0.48 | 0.13 |
> | VLM-as-a-Judge (GPT4.1) | 0.40 | 0.05 | 0.68 | 0.35 |
> | Ours  (Qwen3VL-4b) | 0.45 | 0.10 | 0.65 | 0.31 |
> | Ours  (Qwen3VL-8b) | 0.67 | 0.38 | 0.71 | 0.42 |
> | Ours (GPT4.1) | **0.81** | **0.63** | **0.83** | **0.66** |

---

> > ### Author Response · Authors · 2025-11-26
> > **Response to Reviewer 5Zma (Part 2)**
> >
> > ### **W4. Limited impact on downstream tasks**
> > We understand the concern that manually collected instructions for 3D scene synthesis may appear to have limited impact on the research community. However, we respectfully argue that LEGO-Bench addresses a foundational and underexplored problem that has broad implications for downstream tasks:
> >
> > * **Reveals the limitations of existing text-to-3D scene methods.** LEGO-Bench reveals that existing LLM-based methods achieve under 10% accuracy in generating scenes that match fine-grained instructions. This exposes a fundamental limitation of current LLM-based scene generation and suggests a potential avenue for future research.
> > * **Enables realistic scene generation for embodied tasks.** Prior work has shown that training embodied agents in realistic scenes yields greater performance than in large-scale, unrealistic synthetic environments \[6, 7\]. To address this need for realism, LEGO-Bench provides fine-grained evaluation of text-guided 3D synthesis. This facilitates progress toward generating large-scale, realistic environments, and ultimately enables effective training of generalizable agents across embodied tasks such as robot planning, navigation, and simulation.
> >
> > **References**
> > \[1\] Poole, Ben, et al. "Dreamfusion: Text-to-3d using 2d diffusion." *arXiv preprint arXiv:2209.14988* (2022)., https://arxiv.org/abs/2209.14988
> > \[2\] Duggal, Shivam, et al. "Eval3D: Interpretable and fine-grained evaluation for 3D generation." *Proceedings of the Computer Vision and Pattern Recognition Conference*. 2025., https://arxiv.org/abs/2504.18509
> > \[3\] Lin, Chenguo, and M. U. Yadong. "InstructScene: Instruction-Driven 3D Indoor Scene Synthesis with Semantic Graph Prior." *The Twelfth International Conference on Learning Representations*. 2023., https://arxiv.org/abs/2402.04717
> > \[4\] Jia, Baoxiong, et al. "Sceneverse: Scaling 3d vision-language learning for grounded scene understanding." *European Conference on Computer Vision*. Cham: Springer Nature Switzerland, 2024., https://arxiv.org/abs/2401.09340
> > \[5\] Tam, Hou In Ivan, et al. "SceneEval: Evaluating semantic coherence in text-conditioned 3D indoor scene synthesis." *arXiv preprint arXiv:2503.14756* (2025)., https://arxiv.org/abs/2503.14756
> > \[6\] Deitke, Matt, et al. "🏘️ ProcTHOR: Large-Scale Embodied AI Using Procedural Generation." *Advances in Neural Information Processing Systems* 35 (2022): 5982-5994., https://arxiv.org/abs/2206.06994
> > \[7\] Khanna, Mukul, et al. "Habitat synthetic scenes dataset (hssd-200): An analysis of 3d scene scale and realism tradeoffs for objectgoal navigation." *Proceedings of the IEEE/CVF Conference on Computer Vision and Pattern Recognition*. 2024., https://arxiv.org/abs/2306.11290

---

### Public Comment · ~Hou_In_Ivan_Tam1 · 2025-11-18

Hello authors, thank you for putting together this interesting submission. It is nice to see more work focusing on the evaluation gaps in this area. As I was reading, I had a few questions about both the conceptual choices and the implementation details.

1. I am curious how much the tool-based approach actually contributes. In Appendix C3, some of the tools handle very simple lookups, for example retrieving the set of objects in a room. Could these be replaced with a single dictionary that contains all of this information about the scene, which recent LLMs should be able to parse directly, avoiding the need for extensive runtime tool selection?

2. In Section 4.1.3 you note that text can capture fine-grained attributes, for example the color of small objects, and can also summarize information extracted from retrieved images. If I understand correctly, in LEGO-Eval a VLM first looks at rendered object images to produce this text, and then an LLM or VLM processes this text again downstream. In that case, is the intermediate textual step actually necessary, or could the system achieve similar performance by working directly with the images instead of converting them into text first, which would also reduce the amount of tool use?

3. For relationships like left and right, the frame of reference can be ambiguous. How do you define the coordinate system or viewpoint when evaluating these relations?

4. The textual descriptions in LEGO-Bench seem quite templated, for example sentences like "there is ... in the room" or "\<object\> is or has \<attribute\>". Is this style used throughout the dataset, and do the instructions in Figures 2 and 8 also follow a fixed order, such as rooms first, then objects, then relations in separate sentences? This seems easier to parse than natural descriptions of indoor scenes. Have you considered or tested more free-form descriptions, and how do you think that would affect the results in Table 4?

5. In Section 4.2.1, could you clarify how you adapted I-Design, LayoutGPT, and LayoutVLM to work with Holodeck? On lines 361 to 362, you mention that LayoutGPT and LayoutVLM position a given set of objects, but both methods can also generate scenes from scratch. LayoutVLM provides the prompts they used to select objects in their supplementary material, and the LayoutGPT paper specifies that it is given the available categories and then decides what to place, rather than taking an explicit object list. How did you configure these methods in your experiments to fit the Holodeck setup?

Once again, it is encouraging to see more attention on the evaluation gap in this area. Thank you for the interesting work!

---

> ### Author Response · Authors · 2025-11-29
> **Response to Hou In Ivan Tam (part 1)**
>
> Dear Hou In Ivan Tam,
> Thank you for your interest in our work. We will respond to your questions in the comments below:
>
> ### **Q1. Contribution of tools on rigorous evaluation**
> We appreciate your question regarding the contribution of our tool-based evaluation framework. To better understand the value of our approach, we implemented two new baselines: (1) providing only the scene graph to VLM-as-a-Judge, and (2) providing both the scene graph and the corresponding scene images to VLM-as-a-Judge.
>
> * **Setup.** The outputs of LLM-based 3D scene synthesis methods are used as the scene graph, as they output scenes in scene graph structure in text. The scene images are taken from 5 different perspectives of the scene. We use GPT-4.1 as the base model for all methods.
>
> * **Results.** The results reveal that even when VLM-as-a-Judge had access to both the scene graph and the scene image, LEGO-Eval still outperformed these baselines. A possible explanation for the performance drop is the limitation of large language models in processing long contexts. Additionally, the excessive presence of irrelevant scene information may prevent the model from focusing on the specific details that are truly necessary for accurate evaluation.
>
> LEGO-Eval leverages a diverse set of tools, including those that perform simple lookups, to selectively extract only the information necessary for scene evaluation. Moreover, the outputs of simple tools are used to determine the essential arguments required by more complex tools. This structured and targeted approach allows LEGO-Eval to achieve more accurate evaluations.
>
> The detailed result can be found in the table below:
>
> | Methods | Holistic ↑ |  | Partial ↑ |  |
> | :---: | :---: | :---: | :---: | :---: |
> |  | **F1** | **Cohen’s Kappa** | **F1** | **Cohen’s Kappa** |
> | Scene Images | 0.40 | 0.05 | 0.68 | 0.35 |
> | SceneGraph | 0.37 | 0.04 | 0.66 | 0.36 |
> | SceneGraph \+ Scene Images  | 0.37 | 0.04 | 0.65 | 0.35 |
> | **LEGO-Eval (Ours)** | **0.81** | **0.63** | **0.83** | **0.66** |
>
> ### **Q2. Necessity of multimodal tools**
>
> We find that providing images directly to the VLM, without any textual descriptions, can lead to incorrect focus or misinterpretation of visual content—especially when the model struggles to identify key objects or their attributes. To address this, we introduce multimodal tools such as 'get\_object\_match' and 'get\_property\_description', which either identify the object depicted in the image or extract information relevant to the evaluation constraints. These tools help direct the model’s attention to the appropriate visual elements and result in more accurate assessments.
>
> We empirically validate the necessity of this intermediate textual step in the ablation study (Section 4.1.3), where removing these tools results in a performance drop: Holistic F1 decreases by 0.04%, and Partial F1 drops by 1.02%. Although the absolute difference may appear small, this degradation reflects consistent failures in fine-grained understanding. Therefore, we include these tools to ensure more accurate and reliable evaluation.
>
> ### **Q3. Ambiguity in instructions**
> Thank you for pointing this out. To avoid ambiguity, we explicitly define the viewpoint in the instructions. For example, when the fridge is against the wall, we use phrases like “facing the fridge” so that “left” and “right” are understood from that perspective.

---

> > ### Author Response · Authors · 2025-11-29
> > **Response to Hou In Ivan Tam (part 2)**
> >
> > ### **Q4. Instruction parsing**
> > **Clarification on fixed template of instructions**
> > We would like to clarify that the textual instructions in LEGO-Bench do not follow a fixed template or prescribed order of constraints. During data collection, annotators were not provided with any templates or guidelines regarding constraint ordering.
> >
> > **Constraint identification on free-form instructions**
> > To evaluate whether LEGO-Eval can handle more free-form instructions, we conducted additional experiments where instructions were rephrased using a large language model to introduce variation in phrasing and constraint order. We then used LEGO-Eval to evaluate the four 3D scene synthesis methods with the rephrased instructions.
> >
> > The results shown in the table below reveal that there are only minimal differences compared to evaluations based on the original human-annotated constraints. This indicates that LEGO-Eval is robust to variations in natural language instructions, including free-form instructions. These findings support the claim that LEGO-Eval is not dependent on rigid or templated instruction formats.
> >
> > Detailed results can be found in the table below:
> >
> > | Methods | Holistic |  |  |  | Partial |  |  |  |
> > | :---: | :---: | :---: | :---: | :---: | :---: | :---: | :---: | :---: |
> > |  | M1 | M2 | M3 | M4 | M1 | M2 | M3 | M4 |
> > | Oracle Constraint | 0.12 | 0.05 | 0.05 | 0.14 | 0.66 | 0.57 | 0.35 | 0.65 |
> > | Identified Constraint (Rephrased) | 0.11 | 0.03 | 0.07 | 0.10 | 0.65 | 0.57 | 0.36 | 0.65 |
> > | **Difference in SR** | **\-0.01** | **\-0.02** | **\+0.02** | **\-0.04** | **\-0.01** | **\+0.00** | **\+0.01** | **\+0.00** |
> >
> > ### **Q5. Adapting Holodeck with baseline methods**
> > **Scene generation**
> > We clarify that I-Design \[1\], LayoutVLM \[2\], and LayoutGPT \[3\] do not generate room structures such as walls or floors. Therefore, we use Holodeck \[4\] to generate the basic room layout (walls and floor), and the compared methods are used solely for object selection or/and placement within each room.
> >
> > **Object selection**
> >
> > * **LayoutVLM.** While the supplementary material provides prompts for object selection, these were intended for test data generation and are not part of the LayoutVLM framework itself. Furthermore, LayoutVLM requires a human-in-the-loop for object selection, which makes it incompatible with automated benchmarking. Accordingly, we did not use those prompts; instead, we provided the model with object names and asset UIDs selected by Holodeck.
> >
> > * **LayoutGPT.** Although LayoutGPT selects which objects to place based on a prompt, it is limited to a small, fixed set of 21 categories. In contrast, LEGO-Bench provides access to 51.7K object categories, which are too many to include in a single prompt. To address this, we include only the categories of objects selected by Holodeck in the prompt, allowing LayoutGPT to choose among them.
> >
> > We acknowledge that the description in the manuscript may have caused confusion, and we will revise the text to clarify how we adapt Holodeck with the baseline methods.
> >
> > **References**
> > \[1\] Çelen, Ata, et al. "I-design: Personalized llm interior designer." *European Conference on Computer Vision*. Cham: Springer Nature Switzerland, 2024., https://arxiv.org/abs/2404.02838
> > \[2\] Sun, Fan-Yun, et al. "Layoutvlm: Differentiable optimization of 3d layout via vision-language models." *Proceedings of the Computer Vision and Pattern Recognition Conference*. 2025., https://arxiv.org/abs/2412.02193
> > \[3\] Feng, Weixi, et al. "Layoutgpt: Compositional visual planning and generation with large language models." *Advances in Neural Information Processing Systems* 36 (2023): 18225-18250., https://arxiv.org/abs/2305.15393
> > \[4\] Yang, Yue, et al. "Holodeck: Language guided generation of 3d embodied ai environments." *Proceedings of the IEEE/CVF Conference on Computer Vision and Pattern Recognition*. 2024., https://arxiv.org/abs/2312.09067

---

### Author Response · Authors · 2025-11-26
**General Response**

We sincerely appreciate the reviewers’ thoughtful and constructive feedback. During the discussion period, we devoted our efforts to conducting additional experiments and analyses to address the key suggestions raised.
In this general response, we present the results of the experiments that were commonly requested across the reviews.

## **Cost Tradeoff Comparison**
We appreciate the reviewer’s question regarding the computational tradeoffs between LEGO-Eval and the single-pass VLM-as-a-Judge baseline. The comparison between the methods can be summarized as:

* **Total Time.** A single-pass VLM judge using GPT-4.1 evaluates the full benchmark in 0.5 hours at a cost of \$3, whereas LEGO-Eval with GPT-4.1 requires 2 hours at a higher cost due to multi-step tool calls.
* **Cost.** Both methods with Qwen3VL-32B incur $0 cost. While VLM-as-a-Judge takes shorter time for evaluation, LEGO-Eval achieves partial F1 and Cohen’s kappa similar to that of LEGO-Eval with GPT-4.1.
* **Human Alignment.** LEGO-Eval with both GPT-4.1 and Qwen3VL-32B marginally outperforms VLM-as-a-Judge. Particularly, the difference of Cohen's kappa (holistic) between both methods is more than an order of magnitude.

Thus, if minimal evaluation time is the priority, the single-pass VLM judge is preferable. However, when accuracy or zero-cost evaluation with open models is important, LEGO-Eval consistently provides substantially higher agreement with human judgments.

The detailed cost tradeoff comparison can be found in the table below:

|  Method |  Total Time (LEGO-Bench) ↓ |  Cost (LEGO-Bench) ↓ | Holistic ↑ |  | Partial ↑ |  |
| ----- | :---: | :---: | :---: | :---: | :---: | :---: |
|  |  |  | **F1** | **Cohen’s Kappa** | **F1** | **Cohen’s Kappa** |
| VLM-as-a-judge (Qwen3VL-32B) | 1.5 hours | **$0** | 0.40 | 0.04 | 0.58 | 0.23 |
| VLM-as-a-judge (GPT4.1) | **0.5 hours** | $3 | 0.40 | 0.05 | 0.68 | 0.35 |
| Ours (Qwen3VL-32B) | 4 hours | **$0** | 0.67 | 0.40 | 0.80 | 0.61 |
| Ours (GPT4.1) | 2 hours | $70 | **0.81** | **0.63** | **0.83** | **0.66** |

## **Simulator & Scene Graph Dependency**
We appreciate the reviewers’ concern regarding LEGO-Eval’s dependency on the simulator and scene graph access.

**Clarification on scene graph dependency**

We would like to clarify that LEGO-Eval is primarily designed to evaluate LLM-based methods that themselves generate structured scene representations such as scene graphs. In this sense, the assumption of scene graph access aligns with the typical output format of the target methods, rather than being an unrealistic constraint.

**Clarification on simulator dependency**

We respectfully clarify that LEGO-Eval is not designed to rely on a specific simulator or simulation environment. Instead, the framework is built to support rigorous instruction-scene alignment evaluation by flexibly integrating diverse tools appropriate to the evaluation context.

LEGO-Eval is capable of seamlessly incorporating newly available tools into its evaluation pipeline, as long as they are accessible via tool calls. This modular and simulator-agnostic design enables the framework to be adapted to a wide range of environments—regardless of whether direct access to a particular simulator is available—by allowing users to compose suitable toolsets for each evaluation setting.

---

### Author Response · Authors · 2025-12-01
**Author Final Remarks**

Dear AC and reviewers,

Thank you for your valuable feedback and thoughtful reviews. In particular, we sincerely appreciate the AC’s extra effort in carefully evaluating our submission under this year’s limited discussion process.

We summarize our key responses and clarifications as follows:

---

### **Tool-based evaluation framework contributes to rigorous evaluations.**

* LEGO-Eval outperforms all additional baselines, including those that use scene images annotated with detected instruction-relevant elements \[1\], and those that take as input scene graphs \[2, 3\] or point clouds \[4, 5\] representing the generated scene. \[reviewer zAu4, public comment\]
* With a small base model (Qwen3-VL-8B), LEGO-Eval still outperforms all baselines, demonstrating robustness in resource-constrained settings. \[reviewer 5Zma\]
* Failure mode analysis shows that LEGO-Eval makes fewer errors in identifying and locating scene elements described in instructions, which is enabled by leveraging a diverse set of tools. \[reviewer zAu4, MKU2\]
* LEGO-Eval does not rely on any specific simulator; its modular design enables flexible integration of tools via tool calls, supporting diverse evaluation environments. \[reviewer MKU2, oMe6\]

### **LEGO-Bench supports advances in LLM-based 3D scene synthesis.**

* LEGO-Bench includes 130 instruction-scene pairs, comparable in scale to other impactful text-to-3D benchmarks such as DreamFusion \[6\] with 153 instructions and Eval3D \[7\] with 160\. \[reviewer 5Zma\]
* Prior datasets lack aligned scenes \[8\] or provide them in mesh/point cloud formats unusable by LLM-based methods \[9, 10\]. LEGO-Bench offers scenes in structured text format, enabling compatibility with LLM-based approaches. \[reviewer 5Zma\]
* Previous benchmarks do not provide reliable automatic evaluation \[8, 9, 10\]. We propose LEGO-Eval to enable scalable, automatic alignment assessment between scenes and instructions. \[reviewer 5Zma\]
* LEGO-Bench reveals that current LLM-based methods achieve less than 10% accuracy in aligning generated scenes with fine-grained instructions. This underscores the limitations of existing approaches and provides a concrete benchmark to guide progress in instruction-conditioned 3D synthesis. \[reviewer 5Zma\]

**References**
\[1\] Liu, Shilong, et al. "Grounding dino: Marrying dino with grounded pre-training for open-set object detection." *European conference on computer vision*. Cham: Springer Nature Switzerland, 2024., https://arxiv.org/abs/2303.05499
\[2\] Gu, Qiao, et al. "Conceptgraphs: Open-vocabulary 3d scene graphs for perception and planning." *2024 IEEE International Conference on Robotics and Automation (ICRA)*. IEEE, 2024., https://arxiv.org/abs/2309.16650
\[3\] Huang, Jiani, et al. "Laser: A neuro-symbolic framework for learning spatial-temporal scene graphs with weak supervision." *arXiv preprint arXiv:2304.07647* (2023)., https://arxiv.org/abs/2304.07647
\[4\] Hong, Yining, et al. "3d-llm: Injecting the 3d world into large language models." *Advances in Neural Information Processing Systems* 36 (2023): 20482-20494., https://arxiv.org/abs/2307.12981
\[5\] Guo, Ziyu, et al. "Point-bind & point-llm: Aligning point cloud with multi-modality for 3d understanding, generation, and instruction following." *arXiv preprint arXiv:2309.00615* (2023)., [https://arxiv.org/abs/2309.00615](https://arxiv.org/abs/2309.00615)
\[6\] Poole, Ben, et al. "Dreamfusion: Text-to-3d using 2d diffusion." *arXiv preprint arXiv:2209.14988* (2022)., https://arxiv.org/abs/2209.14988
\[7\] Duggal, Shivam, et al. "Eval3D: Interpretable and fine-grained evaluation for 3D generation." *Proceedings of the Computer Vision and Pattern Recognition Conference*. 2025., [https://arxiv.org/abs/2504.18509](https://arxiv.org/abs/2504.18509)
\[8\] Tam, Hou In Ivan, et al. "SceneEval: Evaluating semantic coherence in text-conditioned 3D indoor scene synthesis." *arXiv preprint arXiv:2503.14756* (2025)., https://arxiv.org/abs/2503.14756
\[9\] Lin, Chenguo, and M. U. Yadong. "InstructScene: Instruction-Driven 3D Indoor Scene Synthesis with Semantic Graph Prior." *The Twelfth International Conference on Learning Representations*. 2023., https://arxiv.org/abs/2402.04717
\[10\] Jia, Baoxiong, et al. "Sceneverse: Scaling 3d vision-language learning for grounded scene understanding." *European Conference on Computer Vision*. Cham: Springer Nature Switzerland, 2024., https://arxiv.org/abs/2401.09340

---

### Meta-Review · Area_Chair_qkfy · 2026-01-07

**Summary:**

While reviewers praised the novelty of the tool-augmented evaluation (LEGO-Eval) over standard VLM judges, the paper falls short of the acceptance threshold due to concerns regarding scalability and scope. The primary friction points were the framework's strict dependency on simulator metadata (making it unusable for mesh/diffusion-based methods) and the limited size of the benchmark.

**Reviewer Concerns:**

Addressed:
Unfair Baselines: The authors successfully added Grounding DINO, ConceptGraphs, and 3D-LLM baselines to prove LEGO-Eval's superiority.
Model Reliance: Authors rebutted that the framework works with open-weights models (Qwen), not just GPT-4.
Cost/Failure Analysis: Detailed cost tradeoffs and error breakdowns were provided.

Outstanding:
Simulator Dependency: The requirement for ground-truth scene graphs restricts utility to specific pipelines (like Holodeck), limiting broader application.
Downstream Evidence: Lack of empirical experiments demonstrating that this specific alignment improves downstream robot learning tasks.

**Reviewer Scores:**

Reviewer 5Zma (2): The reviewer might improve their rating slightly (to 3) due to the "open model" fix, but the fundamental concern that the benchmark is too small to be impactful remains unresolved.

Reviewer zAu4 (4): The author directly addressed this reviewer's request for baselines and cost analysis, which likely results in an improved score of 6.

Reviewer MKU2 (6): The failure mode analysis addressed some concerns, though the simulator constraint prevents a higher score.

Reviewer oMe6 (6): The score likely remains static as the "usability" concern regarding dense annotations was clarified but confirmed as a limitation.

This leads to final likely scores of 6,6,6/4,3 assuming score improvements had the discussion continued.

---

### Decision · Program_Chairs · 2026-01-26

Reject